# Comparison of observed and modelled cloud-free longwave downward radiation (2010-2016) at the high mountain BSRN Izaña station

Rosa Delia García[1,2,3], Africa Barreto[4,2,3], Emilio Cuevas[2], Julian Gröbner[5], Omaira Elena García[2], Angel Gómez-Peláez[2,a], Pedro Miguel Romero-Campos[2], Alberto Redondas[2], Victoria Eugenia Cachorro[3], and Ramon Ramos[2]

[1]Air Liquide España, Delegación Canarias, Candelaria, 38509, Spain
[2]Izaña Atmospheric Research Center (IARC), State Meteorological Agency (AEMET), Spain
[3]Atmospheric Optics Group, Valladolid University, Valladolid, Spain
[4]Cimel Ele ctronique, Paris, France.
[5]Physikalisch-Meteorologisches Observatorium Davos, World Radiation Center (PMOD/WRC), Davos, Switzerland.
[a]now at: Meteorological State Agency of Spain (AEMET), Delegation in Asturias, Oviedo, Spain.

*Correspondence to:* Emilio Cuevas
(ecuevasa@aemet.es)

**Abstract.**

A 7-year (2010-2016) comparison study between measured and simulated longwave downward radiation (LDR) under cloud-free conditions has been performed at the Izaña Atmospheric Observatory (IZO, Spain). This analysis encompasses a total of 2062 cases distributed almost 50 % between day and night. Results show an excellent agreement between Baseline Surface Radiation Network (BSRN) measurements and simulations with libRadtran V2.0.1 and MODTRAN V6 radiative transfer models (RTM). Mean bias (simulated-measured) < 1.1 %, and root mean square of the bias (RMS) < 1 %, are within the instrumental error (2 %). These results highlight the good agreement between the two RTMs, demonstrating to be useful tools for the quality control of LDR observations and for detecting temporal drifts in field instruments. The standard deviations of the residuals, associated to the RTM input parameters uncertainties are rather small, 0.47 % and 0.49 % for libRadtran and MODTRAN, respectively, at day-time, and 0.49 % to 0.51 % at night-time. For precipitable water vapor (PWV) > 10 mm, the observed night-time difference between models and measurements is +5 W m$^{-2}$ indicating a scale change of the World Infrared Standard Group of Pyrgeometers (WISG), which serves as reference for atmospheric longwave radiation measurements. Preliminary results suggest a possible impact of dust aerosol on infrared radiation during day-time that might not be correctly parametrized by the models, resulting in a slight underestimation of the modeled LDR, of about -3 W m$^{-2}$, for relatively high aerosol optical depth (AOD > 0.20).

## 1   Introduction

Longwave downward radiation (LDR) at the Earth's surface is a key component in land-atmosphere interaction processes, and is crucial in surface energy budget and global climate change, because the changes in the LDR values may be related to changes

in cloud-cover, cloud type, water vapour, temperature, and the increase of anthropogenic greenhouse gas concentrations in the atmosphere (Wild et al., 1997; Marty et al., 2003; Iacono et al., 2008; Philipona et al., 2012; Wild et al., 2013; Wang and Dickinson, 2013; Wild et al., 2015). Thus, LDR measurements and simulations are needed to understand the processes involved in the changes on the LDR sources and levels, and their possible relations with the sources of climate change (Dutton, 1993; Wild et al., 2001).

Atmospheric longwave irradiance measurements are usually performed with hemispherical receivers on flat horizontal surfaces. The LDR is mainly measured with pyrgeometers, with the Eppley Precision Infrared Radiometer (PIR), EKO MS-201 Precision Pyrgeometer, and Kipp and Zonen CG(R) series being the most used (McArthur, 2005). These latter pyrgeometers have been designed for LDR measurements with high reliability and accuracy. The estimated uncertainty for LDR instantaneous values, indicated by the Baseline Surface Radiation Network (BSRN) in 2004, is 3 W m$^{-2}$ (2 %) (McArthur, 2005). These values account for calibration uncertainties and are estimated from standard deviation of the calibration coefficients. The Baseline Surface Radiation Network (BSRN) accuracy target for LDR is $\pm$ 2 W m$^{-2}$ and the average observed LDR change from 24 BSRN sites since early 1990s has been +2 W m$^{-2}$ dec$^{-1}$ (Wild, 2017) as a result of the increase of the greenhouse effect.

At the beginning of the $20^{th}$ century, several methods and equations were developed to estimate LDR when or where no measurements were available. The first parameterization of the LDR was developed by Ångström (Ångström, 1918), who developed an empirical relationship between cloud-free emissivity and water vapour pressure at the surface. Following the pioneer work of Ångström, several authors (e.g. Brunt (1932); Swinbank (1963); Idso and Jackson (1969); Brutsaert (1975); Prata (1996)) proposed diverse relationships capable of simulating LDR based on relations between LDR, vapour pressure, temperature and the Stefan-Boltzmann constant, since the theoretical basis of this parameterization is the assumption that the atmosphere behaves as a grey body:

$$LDR = \epsilon(T,e)\sigma T^4 \tag{1}$$

In this equation $\epsilon(T,e)$ is the cloud-free atmospheric emissivity, $T$ and $e$ are the air temperature and the water vapor pressure measured at the surface, respectively, and $\sigma$ is the Stefan-Boltzmann constant (5.67x10$^{-8}$ W m$^{-2}$ K$^{-4}$). The above mentioned parameterizations show uncertainties ranging from 9 % to 15 % in low altitude sites while at high altitude sites the LDR estimations present uncertainties ranging from 12 % to 21 %. More recently, Iziomon et al. (2003) presented an improved parameterization that reduces the uncertainties to 6 % for lowland sites and 7 % for mountain sites for all-sky conditions. Ruckstuhl et al. (2007) showed that the monthly mean LDR can be effectively modelled from specific humidity or water vapour obtaining differences < 5 %. Dupont et al. (2008) presented a more sophisticated parameterization based on the vertical profiles of temperature and humidity obtaining uncertainties of ~ 5 W m$^{-2}$ for cloud-free conditions, for both day-time and night-time.

The need to provide more accurate LDR estimates from models to improve climate forecasting, led to the introduction of Radiative transfer models (RTMs) adapted or developed to simulate such LDRs. There exist several studies in the literature aiming to compare measured and simulated LDR (Morcrette, 2002; Dürr et al., 2005; Marty et al., 2003; Long and Turner,

2008; Wacker et al., 2011; Viúdez-Mora et al., 2009; Viúdez-Mora et al., 2015). The key point in these studies is the use, of data from radio soundings launched at the measurement site which provide vertical profiles of humidity, pressure and temperature, as model inputs.

An intercomparison performed by Schweizer and Gautier (1995) with LOWTRAN model under cloud-free conditions
showed that the model simulations generally exceed measured LDR values with a bias of -0.7 $\pm$ 11 W m$^{-2}$ and a root mean square error (RMSE) of 10.6 W m$^{-2}$ (4 % of the measured values). In a similar study, Viúdez-Mora et al. (2009) compared LDR measurements and simulations, under cloud-free conditions, with Santa Barbara Disort Atmospheric Radiative Transfer (SBDART; (Ricchiazzi et al., 1998)) at two different sites, Payerne (Switzerland) and Gerona (Spain) obtaining differences of -2.7 $\pm$ 3.4 W m$^{-2}$ and 0.3 $\pm$ 9.4 W m$^{-2}$, respectively. Dürr et al. (2005) found a good agreement between LDR measurements
and simulations with the MODerate resolution atmospheric TRANsmission model (MODTRAN; Berk et al. (2000)), with values of +1.5 W m$^{-2}$ and -3.2 W m$^{-2}$ for night-time (274 cases) and day-time (94 cases), respectively, at Payerne station.

The main goal of this work is to compare BSRN LDR measurements with simulations made with two complex models using observed and modelled data from a relatively long period (between 2010 and 2016). The Izaña Atmospheric Observatory (IZO, http://izana.aemet.es) is an optimal station to carry out this study, because all the model input parameters (precipitable water
vapor (PWV), aerosol optical depth (AOD), total ozone, $N_2O$ in situ, $CO_2$ in situ, $CO_2$ profile and meteorological radiosondes) are measured at the station. This work is divided into six sections. Sect. 2 describes the main characteristics of the IZO test site. In Sect. 3 the technical description of instruments and measurements performed at IZO are shown, as well as the method used for the detection of cloud-free days. Sect. 4 introduces the libRadtran and MODTRAN models and the model input parameters used in this work as well as a theoretical uncertainty assessment of the simulations made with both models. The results of the
comparison and the temporal stability of the LDR observations are shown in Sect. 5, and finally, the summary and conclusions are given in Sect. 6.

## 2   Site Description

The Izaña Atmospheric Observatory is a high-mountain observatory located in Tenerife (Canary Islands, Spain at 28.3° N, 16.5° W, 2373 m a.s.l.). IZO is managed by the Izaña Atmospheric Research Center (IARC) which forms part of the Me-
teorological State Agency of Spain (AEMET). Its location in the Atlantic Ocean and above a stable inversion layer, typical for subtropical regions, provides clean air and clear sky conditions most of the year, offering excellent conditions for calibration and validation activities. In 1984, IZO became a member of the World Meteorological Organization (WMO) Background Atmospheric Pollution Monitoring Network (BAPMoN) and in 1989 it became a Global Atmosphere Watch (GAW) station. In addition, it has been actively contributing to international radiation networks and databases such as NDACC
(Network for the Detection of Atmospheric Composition Change; http://www.ndsc.ncep.noaa.gov/) since 1999, AERONET (Aerosol Robotic Network; http://aeronet.gsfc.nasa.gov/) since 2004, TCCON (Total Carbon Column Observing Network; http://www.tccon.caltech.edu/) since 2007 and the BSRN since 2009, among others. Moreover, since 2014, IZO was appointed by WMO as a CIMO (Commission for Instruments and Methods of Observation) Testbed for aerosols and water vapor remote

**Table 1.** CG(R)4 pyrgeometers installed at IZO.

| Instrument | C [$\mu$V W$^{-1}$ m$^{-2}$] | Calibration Date |
|---|---|---|
| CG(R)4 Kipp & Zonen #080022 | $10.37 \pm 0.34$ | February 2008 |
| CG(R)4 Kipp & Zonen #050783 | $9.39 \pm 0.31$ | June 2014 |
| | $9.41 \pm 0.30$ | March 2017 |

sensing instruments (WMO, 2014a). Updated details of the site and the observation programs can be found in Cuevas et al. (2017b).

## 3 Observational Data and Methods

### 3.1 Instrument and Measurements

The LDR measurements used in this study have been performed by the Izaña BSRN (#61, IZA; http://bsrn.aemet.es) (García et al., 2012) with a broadband Kipp & Zonen CG(R)4 pyrgeometer (onwards, CG(R)4) mounted on a sun tracker equipped with dome shading. This instrument uses a specially designed silicon window which provides a $180°$ field of view (although not hemispherical) with good cosine response. A diamond-like surface protects the outer surface of the window, while the inner surface filters most of solar radiation. The design of the instrument is such that solar radiation absorbed by the windows

is conducted away to reduce the solar heating effect. This fact reduces the need for dome heating correction terms and shading from the sun (McArthur, 2005).

In this study, we analyzed measurements performed with two CG(R)4 series (see Table 1) between 2010 and 2016 at IZO. The CG(R)4 #080022 was calibrated by the manufacturer in February 2008 at Holland (Kipp & Zonen) and the CG(R)4 #050783 was calibrated in June 2014 and March 2017 at the Physikalisch-Meteorologisches Observatorium Davos/World

Radiation Center (PMOD/WRC). Given the two calibration coefficients of the second instrument (see Table 1), we estimate that its degradation is very small, lower than 0.08 % /yr.

The World Radiation Monitoring Center (WRMC) recommends performing quality checks to BSRN data attending to physically possible (PP, minimum 40 - maximum 700 W m$^{-2}$) and extremely rare LDR limits (ER, minimum 60 - maximum 500 W m$^{-2}$), as well as considering the comparison between LDR and air temperature (Gilgen et al., 1995; Long and Dutton, 2002).

We have applied these BSRN quality controls to the IZO LDR measurements and found that the LDR measurements are within the above mentioned limits (Figure 1).

### 3.2 Cloud-free detection

The cloud-free days were detected by using the algorithm developed by (Marty and Philipona, 2000). A Clear-Sky Index (CSI) is calculated to separate cloud-free days from cloudy days using accurate measurements of LDR in conjunction with air

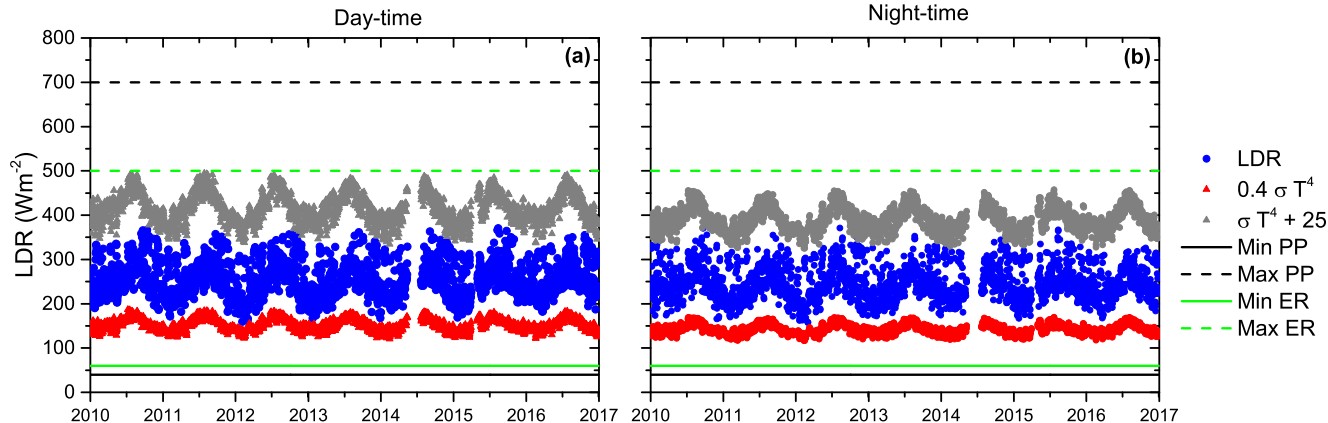

**Figure 1.** The LDR time series obtained at (a) day-time and (b) night-time with a CG(R)4 pyrgeometer between 2010 and 2016 at IZO BSRN (blue dots). The black and green lines represent the physically possible (Min PP, Max PP) and extremely rare limits (Min ER, Max ER), respectively and the grey and red dots represent the upper ($\sigma\,T^4$ + 25) and lower (0.4 $\sigma\,T^4$) limit, respectively, where $\sigma$ is Stephan-Boltzmann constant (5.67x10$^{-8}$ W m$^{-2}$ K$^{-4}$) and T is the air temperature in K.

temperature and relative humidity values measured at the station. The CSI index is defined as:

$$CSI = \epsilon_A/\epsilon_{AC} \tag{2}$$

where

$$\epsilon_A = LDR/\sigma T^4 \tag{3}$$

5   and

$$\epsilon_{AC} = \epsilon_{AD} + k(e/T)^{1/8} \tag{4}$$

where $\sigma$ is the Stefan-Boltzmann constant, *T* is air temperature (*K*), $\epsilon_{AD}$ is an altitude-dependent emittance of a completely dry atmosphere, *k* is a location dependent coefficient and *e* is the water vapor pressure (Pa). If CSI Index $\leq$ 1 indicates cloud-free (no clouds) and if CSI Index > 1 indicates cloud-sky (Marty and Philipona, 2000).

10   In order to calculate $\epsilon_{AC}$, this method requires the evaluation of $\epsilon_{AD}$ and *k*, as shown in equation (4). A sample of known cloud-free days is used to plot $\epsilon_{AC}$ against *e/T* (Figure 2). The cloud-free condition of this sample is assured by applying the Long and Ackerman's method (Long and Ackerman (2000); adapted for IZO by García et al. (2014)). This method is based on surface measurements of global and diffuse solar radiation with a 1-min sampling period and consists of in four individual tests applied to normalized global radiation magnitude, maximum diffuse radiation, change in global radiation with time, and

15   normalized diffuse radiation ratio variability. We have considered the period 2010-2016 at 11 UTC to determine the fitting

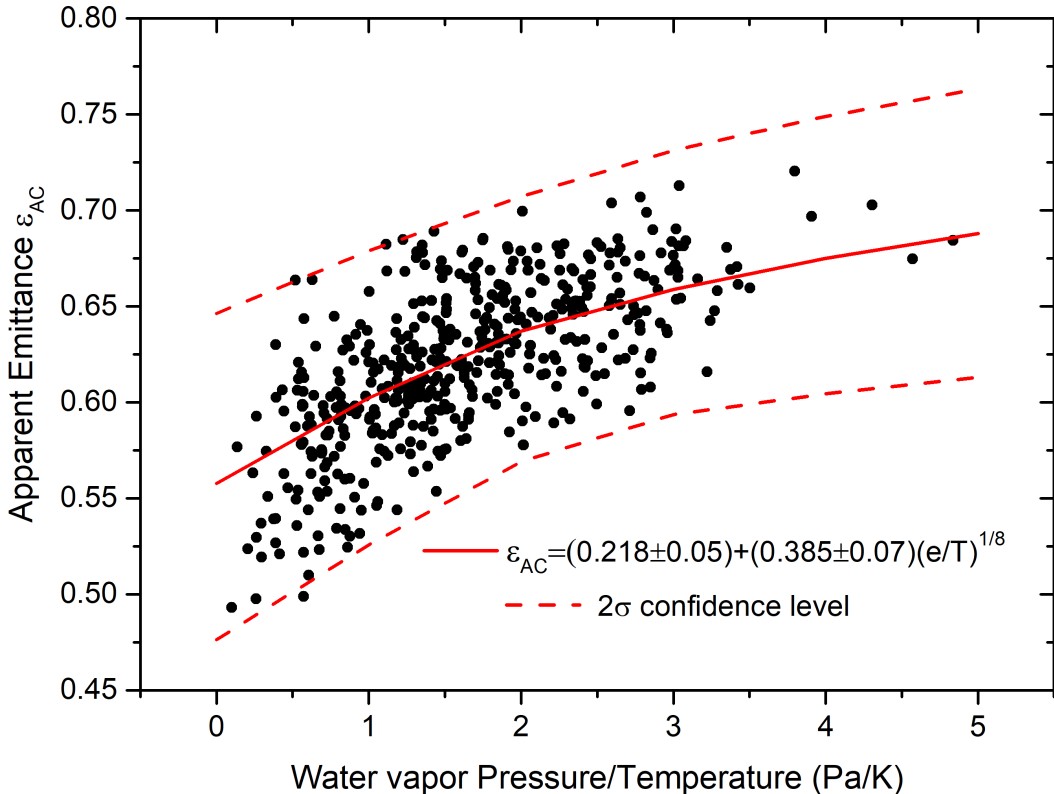

**Figure 2.** Apparent emittance ($\epsilon_{AC}$) as a function of the ratio of screen level water vapor pressure and temperature at IZO in the period 2010-2016 at 11 UTC.

coefficients of equation (4) obtaining the following relationship (Figure 2):

$$\epsilon_{AC} = (0.218 \pm 0.05) + (0.385 \pm 0.07)(e/T)^{1/8} \qquad (5)$$

Despite the $\epsilon_{AD}$ depends on the altitude of the station we have obtained for IZO a value of 0.218, similar to the values obtained by Marty and Philipona (2000) for stations located between 2230 and 2540 m (0.220 and 0.211, respectively).

5     Once we have adjusted the coefficients, the cloud-free cases were selected with a combination of Long and Ackerman and CSI methods. At day-time, we have used the Long and Ackerman method, taking into account the period 11-13 UTC for each day. At night-time the CSI was applied in the period 23-01 UTC. These results have been checked by visual examination of 5-minute total sky images obtained with a SONA camera (Automatic Cloud Observation System; Gónzalez et al. (2012)) running at IZO. We found that both methods detect 97 % of the visually selected cases. A total of 1161 and 1083 cases were

10   detected for day-time and night-time, respectively, in the period 2010-2016.

## 4  Radiative transfer models and input parameters

The simulations of surface LDR were determined with two RTMs: libRadtran and MODTRAN.

The LibRadtran model (freely available from http://www.libradtran.org; Mayer and Kylling (2005)) used in this work is the version 2.0.1 (Emde et al., 2016). The simulations were performed with highly resolved absorption coefficients that were calculated using the absorption band parameterization called REPTRAN. It is based on the HITRAN 2004 spectroscopic database, in which wavelength-integrals have been parameterized as weighted means over representative wavelengths (Gasteiger et al., 2014). The simulations performed using REPTRAN in the thermal range showed relative differences of about 1 % with respect to simulations performed with high spectral resolution models and they are 6-7 times better than the simulations done with the LOWTRAN band parameterization (Gasteiger et al., 2014).

The MODTRAN version used in this work is the MODTRAN v6 (Berk and Hawes, 2017), an atmospheric transmittance and radiance model developed by the U. S. Air Force Research Laboratory in collaboration with Spectral Sciences, Inc. We have selected a band model with a resolution of 1 cm$^{-1}$ for spectral calculations. The MODTRAN band model molecular spectroscopy is based on the HIgh-resolution TRANsmission molecular absorption (HITRAN) database (Rothman et al., 2013).

The main differences between the two models is in the molecular absorption band: while MODTRAN uses HIgh-resolution TRANsmission molecular absorption (HITRAN) database (Rothman et al., 2013), LibRadtran uses the absorption band parameterization called REPTRAN (Gasteiger et al., 2014).

For both models, the LDR simulations were calculated by using as radiative transfer equation (RTE) solver the DISORT (DIScrete ORdinate Radiative Transfer solvers), developed by Chandrasekhar (1960) and Stamnes et al. (1988, 2000), and based on the 5 multi-stream discrete ordinates algorithm. The number of streams used to run Disort was 16. For each simulation, the integrated downward irradiance has been calculated in the spectral range 4-100 $\mu$m.

The two models were run using the same inputs, atmosphere and geometry in order to minimize.

The rest of the inputs measured at IZO are:

– **Radiosondes: Temperature and relative humidity (RH) profiles**

In this work, we have used the AEMET's meteorological radiosondes dataset. Radiosondes are launched twice a day, at 11 and 23 UTC at the Güimar station (WMO GRUAN station #60018, 105 m a.s.l.). This station is located at the coastline, approximately 15 km to the southeast of IZO. Vertical profiles of pressure, temperature and relative humidity were obtained with Vaisala RS92 radiosondes (Rodriguez-Franco and Cuevas, 2013; Carrillo et al., 2016). We have used the radiosonde profiles from the altitude of IZO (2373 m a.s.l.).

– **PWV**

Since January 2009, the PWV has been obtained every 1h at IZO from a (Global Navigation Satellite System (GNSS) receiver considering satellite precise orbits (Romero Campos et al., 2009). In this work, we have used the PWV median measured between 11-13 and 23-01 UTC in order to take into account the radiosonde flight time, and hence making possible a comparison with GNSS observations.

- **$CO_2$ and $N_2O$ profiles**

  The volume mixing ratio (VMR) profiles of atmospheric $CO_2$ and $N_2O$ trace gases were used. These were obtained from the monthly average profiles performed with the ground-based Fourier Transform InfraRed spectrometer (FTIR) at IZO between 1999 and 2015 (Schneider et al., 2005; García et al., 2014; Barthlott et al., 2015). The FTIR program at IZO is part of the Network for the Detection of Atmospheric Composition Change (NDACC). In this study FTIR climatological profiles have been used. The profiles were scaled on a daily basis with ground-level in situ $CO_2$ and $N_2O$ mixing ratios, continuously measured at IZO since June 1984 and June 2007, respectively, within the WMO GAW programme (Cuevas et al., 2017b).

- **$CO_2$ and $N_2O$ in situ**

  Since 2007 the CO2 in situ measurements have been performed with a NDIR analyzer (LICOR-7000) (Gómez-Peláez and Ramos, 2009; Gómez-Peláez et al., 2010) and the N2O in situ measurements with a VARIAN (GC-ECD 3800) (Scheel, 2009). We have used in this work only the night-time (20-08 UTC) averaged $CO_2$ and $N_2O$ data because during this period IZO is under background free troposphere conditions, and the observatory is not affected by local and regional sources of such gases.

- **AOD**

  Atmospheric aerosols have been included in the simulation process by means of the column-integrated aerosol optical depth (AOD), extracted from AERONET (Level 2.0 of version 2, cloud screened and quality ensured). AOD is obtained from solar observations performed with CIMEL sunphotometers at different wavelengths (Holben et al., 1998; Dubovik and King, 2000; Dubovik et al., 2006). The Shetle's aerosol model (Shettle, 1990) has been used in this study. The default properties are: rural type aerosol in the boundary layer, background aerosol above 2 km, spring-summer conditions and a visibility of 50 km. In this work, AOD at 500 nm has been used as model input. For day-time we have used the nearest AOD value to 11 UTC, and for night-time the last available AOD value of the day.

- **Total ozone column (TOC)**

  TOC measurements with Brewer spectrometer began in 1991 at IZO. Since 2003 IZO has been appointed the Regional Brewer Calibration Center for Europe (RBCC-E; http://www.rbcc-e.org) and the total ozone program has been part of NDACC network. We have considered daily total ozone mean value as model input.

## 4.1 Uncertainty due to the input parameters

In this section, we have estimated the theoretical uncertainty for the libRadtran and MODTRAN LDR simulations due to the uncertainties in the input parameters. According to the Guide to the expression of uncertainty in measurement (GUM) (BIPM et al., 2008), we have assumed the Type A uncertainties listed in Table 2.

Our uncertainty estimation is based on two steps: first, the LDR simulations were conducted using the measured values for all the input parameters listed in the previous section, obtaining the non-perturbed values (Sim). In a second step, we

**Table 2.** Assumed Type A uncertainty in the input parameters and their corresponding references

| Uncertainty Source | Standard Uncertainty ($\delta$) | Reference |
|---|---|---|
| **AOD** | $\pm$ 0.01 | Holben et al. (1998); Eck et al. (1999) |
| **TOC** | $\pm$ 1 % | Redondas and Cede (2006) |
| **PWV** | < 3.5 mm: $\pm$ 20 % | Schneider et al. (2010) |
| | $\geq$ 3.5 mm: $\pm$ 10 % | |
| **$N_2O$ in situ** | $\pm$ 0.2 ppbv | Gómez-Peláez and Ramos (2009) |
| **$CO_2$ in situ** | $\pm$ 0.1 ppmv | Zellweger et al. (2015) |
| **$N_2O$ profile** (FITR) | 2.37-20 km: $\sim$ 1 % | García et al. (2016) |
| | > 20 km: 2.0 - 2.5 % | |
| **$CO_2$ profile** (FTIR) | 0.3 % | García et al. (2016) |
| **Temperature profile** | 1080-100 hPa: 0.2° C | |
| | 100-20 hPa: 0.3° C | Vaisala (2013) |
| | 20-3 hPa: 0.5° C | |
| **RH profile** | 2 % | Vaisala (2013) |

have simulated again the same sample but applying the uncertainties listed in Table 2, giving the perturbed values (Sim + $\delta$) (Schneider and Hase, 2008; García et al., 2014). This uncertainty estimation has been applied to those cloud-free days for which all the inputs were available at 11 and 23 UTC between 2010 and 2016 (1048 and 1014 cases at 11 and 23 UTC, respectively). Note that the errors of the FTIR $CO_2$ and $N_2O$ profiles have been theoretically estimated by following the formalism detailed 5 by Rodgers (2000) and assuming the uncertainty sources and values shown in García et al. (2016).

For each uncertainty component we obtain the standard deviation of the measurement residuals from the scatter around the regression line which is related to the correlation coefficient of the least squares fit and the scatter of the perturbed distribution.

The results of the total uncertainty amount analysis are summarized in Table 3. The uncertainties of the PWV and AOD dominate the total uncertainty amount with respect to the other components. The uncertainty of PWV presents a scatter of 10 0.84 W m$^{-2}$ (0.46 %) at day-time, and 0.86 W m$^{-2}$ (0.48 %) at night-time for libRadtran. The results are very similar for MODTRAN, with a scatter of 0.85 W m$^{-2}$ (0.46 %) at day-time and 0.91 W m$^{-2}$ (0.50 %) at night-time. The AOD is also a significant uncertainty source with a scatter of 0.30 W m$^{-2}$ (0.09 %) at day-time and lower scatter at night-time, observing a LDR bias of 1.01 W m$^{-2}$ for libRadtran, and a LDR bias of 0.39 W m$^{-2}$ (0.16 %) at day-time for MODTRAN. The PWV uncertainties present lower scatter at day-time than at night-time, contrary to that observed in the study of the AOD uncertainty 15 for both models. In general, we find that the standard deviations of the LDR residuals are rather small: 0.89 W m$^{-2}$ (0.47 %) and 0.93 W m$^{-2}$ (0.49 %) at day-time, and 0.88 W m$^{-2}$ (0.49 %) and 0.95 W m$^{-2}$ (0.51 %) at night-time, for libRadtran and MODTRAN, respectively.

**Table 3.** Estimation of Type A uncertainties (in W m$^{-2}$ and in % (in brackets)), Sensitivity (%), and Bias (W m$^{-2}$) of the difference between non-perturbed and perturbed LDR simulations (Simulation- (Simulation+ $\delta$)) with libRadtran and MODTRAN models. The combined uncertainty is calculated as the root square sum of all the uncertainty components.

| Uncertainty component | LDR (day-time) | | LDR (night-time) | |
|---|---|---|---|---|
| | STD of the residuals | Regression | STD of residuals | Regression |
| | W m$^{-2}$ (%) | (Sens / Bias) | W m$^{-2}$ (%) | (Sens / Bias) |
| | (Sim + $\delta$) | (%) / W m$^{-2}$ | (Sim + $\delta$) | (%) / W m$^{-2}$ |
| **LibRadtran model** | | | | |
| **AOD** | 0.30 (0.09) | -0.73 / 1.65 | 0.23 (0.08) | -0.46 / 1.01 |
| **TOC (DU)** | <0.01 (<0.01) | <0.01 / <0.01 | <0.01 (<0.01) | <0.01 / <0.01 |
| **PWV (mm)** | 0.84 (0.46) | -1.20 / 6.26 | 0.86 (0.48) | -1.42 / 6.89 |
| **CO$_2$ in situ (ppm)** | <0.01 (<0.01) | <0.01 / <0.01 | <0.01 (<0.01) | <0.01 / <0.01 |
| **N$_2$O in situ (ppb)** | <0.01 (<0.01) | <0.01 / <0.01 | <0.01 (<0.01) | <0.01 / <0.01 |
| **Temperature profile** | <0.01 (<0.01) | <0.01/ <0.01 | 0.03 (<0.01) | <0.01 / <0.01 |
| **RH profile** | <0.01 (<0.01) | <0.01 / <0.01 | <0.01 (<0.01) | <0.01 / <0.01 |
| **Combined uncertainty (u)** | **0.89 (0.47)** | | **0.88 (0.49)** | |
| **MODTRAN model** | | | | |
| **AOD** | 0.39 (0.16) | 0.59 / -1.37 | 0.27 (0.10) | 0.42 / -0.91 |
| **TOC (DU)** | <0.01 (<0.01) | <0.01 / <0.01 | <0.01 (<0.01) | <0.01 / 0.03 |
| **PWV (mm)** | 0.85 (0.46) | -1.18 / 6.19 | 0.91 (0.50) | -1.48 / 7.03 |
| **CO$_2$ in situ (ppm)** | <0.01 (<0.01) | <0.01 / <0.01 | <0.01 (<0.01) | <0.01 / <0.01 |
| **N$_2$O in situ (ppb)** | <0.01 (<0.01) | <0.01 / <0.01 | <0.01 (<0.01) | <0.01 / <0.01 |
| **Temperature profile** | <0.01 (<0.01) | <0.01/ <0.01 | 0.02 (<0.01) | <0.01 / <0.01 |
| **RH profile** | <0.01 (<0.01) | <0.01 / <0.01 | <0.01 (<0.01) | <0.01 / <0.01 |
| **Combined uncertainty (u)** | **0.93 (0.49)** | | **0.95 (0.51)** | |

## 5 Results

### 5.1 BSRN vs Model LDR comparison

In this section, we present the comparison between LDR measured with BSRN and simulated with libRadtran and MODTRAN, considering the available and coincident cloud-free BSRN at day-time and night-time, and the inputs indicated in Sect. 4 at IZO between 2010 and 2016. A total of 1048 measurements at day-time, and 1014 measurements at night-time were used.

The observations were averaged in a time period of 30 minute, in order match the flight time of the radiosonde over IZO. In particular, we averaged over 11:00-11:30 UTC and 23:00-23:30 UTC periods, for day-time and night-time measurements, respectively. The simulations with the two models show an excellent agreement at both day-time (Figure 3a and 3b) and night-

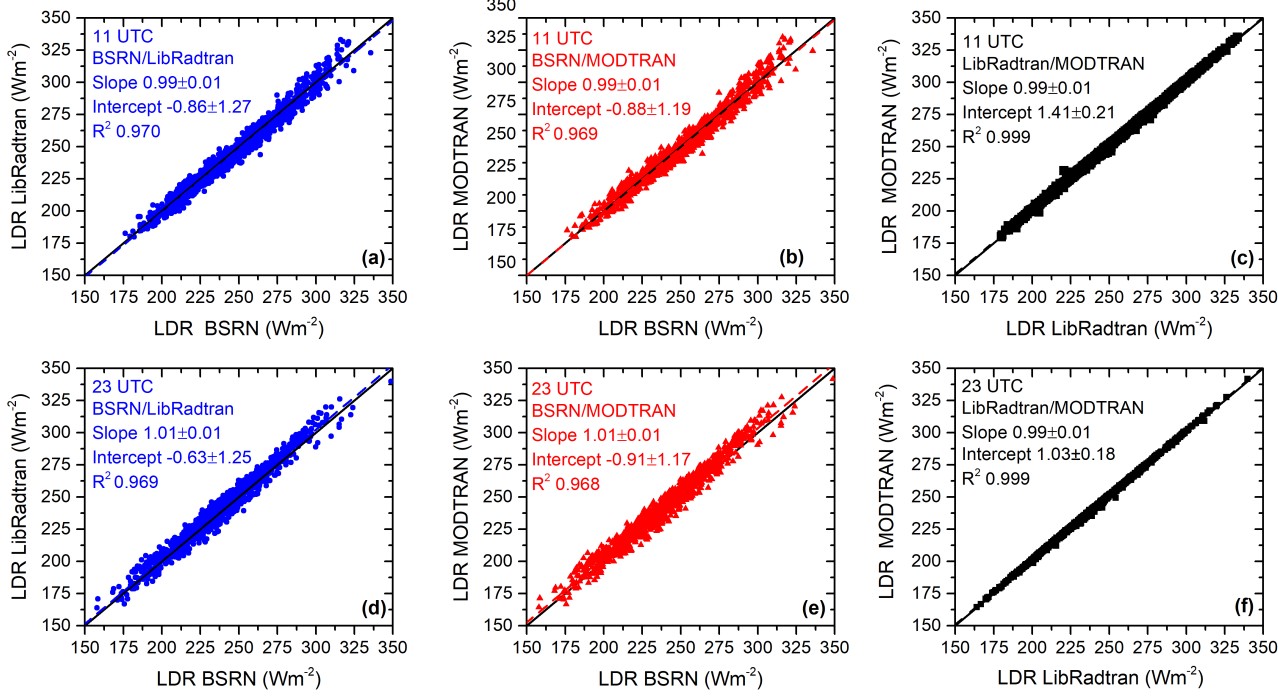

**Figure 3.** Scatterplot of the LDR (W m$^{-2}$) simulations with libRadtran (blue color) versus BSRN LDR (W m$^{-2}$) at cloud-free (a) day-time and (d) night-time. Scatterplot of the MODTRAN LDR (W m$^{-2}$) (red color) versus BSRN LDR (W m$^{-2}$) at (b) day-time and (e) night-time, and scatterplot of the MODTRAN LDR (W m$^{-2}$) (black color) versus libRadtran LDR (W m$^{-2}$) at (c) day-time and (f) night-time. The black solid lines are the diagonal (x=y). The dashed lines represent the least-square fits and the fitting parameters are shown in the legend.

time (Figure 3d and 3e). Both models show a very similar performance, as indicated by the least-square fit with slope of 0.99 and R$^2$ of ∼ 0.970, with a slightly better similitude during the night-time (Figure 3c and 3f).

In order to quantify the difference between BSRN LDR and simulations, we have calculated the absolute LDR difference or bias (simulation-measurement, in W m$^{-2}$), and relative LDR differences ((simulation-measurement)/measurement, in %). As a summary, Table 4 lists the metrics used to quantify these differences.

The results obtained show that both models have a very similar behavior and yield similar performances, as seen in Figures 3c and 3f. Both models underestimate the LDR at day-time between -1.73 W m$^{-2}$(-1.1 %) for BSRN/libRadtran, and -1.79 W m$^{-2}$ (-0.7 %) for BSRN/MODTRAN. In addition, at night-time, both models overestimate with respect to BSRN LDR between 0.15 W m$^{-2}$ (0.1 %) for BSRN/libRadtran, and 1.14 W m$^{-2}$ (0.5 %) for BSRN/MODTRAN. The RMS is < 3 % for both comparisons at day-time, and < 2% at night-time.

The comparison between BSRN LDR and simulations present better results (lower MB, STD and RMS) during night-time than during day-time. These results also agree with other short-term studies. For example, Dürr et al. (2005) found differences between LDR measurements and simulations with MODTRAN of -3.2 W m$^{-2}$ and 1.5 W m$^{-2}$ for day-time (94 cases) and

**Table 4.** Statistics for the LDR bias between libRadtran and MODTRAN simulations and BSRN LDR at IZO (in W m$^{-2}$) performed with data at day-time (1048 cases) and night-time (1014 cases) in the period 2010-2016 (MB, mean bias; RMS, root mean square of the bias and R$^2$). The statistics for the relative bias are in brackets (in %).

| | Day-time | | | Night-time | | |
|---|---|---|---|---|---|---|
| | **MB** | **RMS** | **R**$^2$ | **MB** | **RMS** | **R**$^2$ |
| **BSRN/LibRadtran** | -1.73 | 6.52 | 0.970 | 0.15 | 4.41 | 0.969 |
| | (-1.1%) | (2.6%) | | (0.1%) | (1.8%) | |
| **BSRN/MODTRAN** | -1.79 | 6.30 | 0.969 | 1.14 | 4.53 | 0.968 |
| | (-0.7%) | (2.5%) | | (0.5%) | (1.9%) | |
| **LibRadtran/MODTRAN** | 0.94 | 1.26 | 0.999 | 1.00 | 1.23 | 0.999 |
| | (0.4%) | (0.5%) | | (0.4%) | (0.5%) | |

night-time (274 cases), respectively. Wacker et al. (2009) compared the measurements and simulations with three different models for 39 cloud-free nights in Payerne, finding differences of -1.2 ± 2.5 W m$^{-2}$ with MODTRAN, and 6.0 ± 2.9 W m$^{-2}$ with LOWTRAN. Viúdez-Mora et al. (2009) found differences of -2.7 ± 3.4 W m$^{-2}$ for a total of 44 night-time cases between LDR measurements and simulations with SBDART in Payerne.

According to the results obtained in Sect. 4.1, the uncertainties on PWV and AOD dominate the total uncertainty, thus, the LDR bias have been analyzed.

The box plot of LDR bias for different PWV is presented in Figures 4a and 4b. Both models tend to underestimate LDR (up to 5 W m$^{-2}$) in the case of day-time measurements with PWV < 9 mm (Figure 4a). A LDR bias around zero is observed for higher PWV, although it is necessary to emphasize that the number of data in this PWV range (between 4 % and 5 %) is much

lower. At night-time, the dependence of LDR bias with PWV shows a negligible bias under dry conditions (PWV < 6 mm), and a slight overestimation of both models (up to + 5 W m$^{-2}$) for higher PWV values (Figure 4b). These results are consistent with those obtained by Gröbner et al. (2014) and Nyeki et al. (2017) which argue that the World Infrared Standard Group (WISG) of pyrgeometers has a negative bias of about 5 W m$^{-2}$ under cloud-free conditions and PWV > 10 mm.

Similar results were observed in Figure 4c and 4d, where the dependence of LDR bias with AOD at 500 nm is shown. This

may be due to the fact that PWV and AOD are not completely independent at the Izaña Observatory. In fact these parameters show a moderate correlation (R$^2$ = 0.27 in daytime and R$^2$ = 0.19 in night-time). The reason is that the Saharan Air Layer (SAL) intrusions into the subtropical free troposphere over the North Atlantic are not only associated with dust-laden air masses (higher values of AOD) but also with more content in water vapour (higher PWV) as described by Rodriguez-Franco and Cuevas (2013) and Andrey et al. (2014). The Saharan dust intrusions in the Canary Islands occur the intrusions of Saharan

dust in the Canary Islands have a pulsating character, especially in summer, alternating pristine days with periods of hazy days (Cuevas et al., 2017a).

In order to separate the dependence of LDR bias with PWV from AOD, and viceversa, we have analyzed, on one hand, the LDR bias in function of PWV considering very low aerosols conditions (AOD ≤ 0.05) (Figures 5a and 5b) and, on the

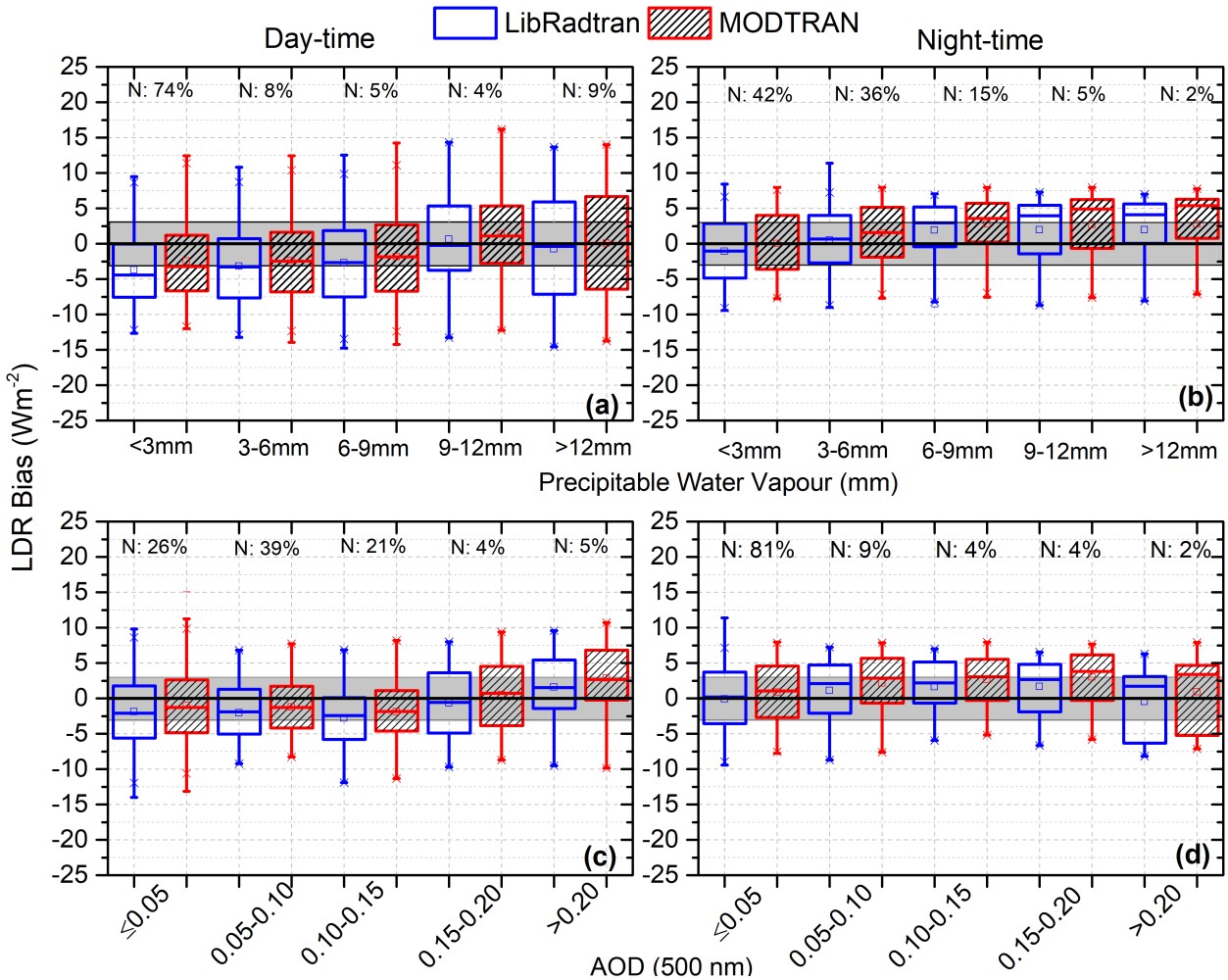

**Figure 4.** Box plot of mean LDR bias (Model-BSRN in W m$^{-2}$) versus PWV (mm) (a) at day-time, (b) at night-time and versus AOD (500 nm) (c) at day-time, (d) at night-time between 2010 and 2016. Lower and upper boundaries for each box are the $25^{th}$ and $75^{th}$ percentiles; the solid line is the median value; the crosses indicate values out of the 1.5 fold box area (outliers); and hyphens are the maximum and minimum values. The blue boxes represent libRadtran/BSRN and the red ones represent MODTRAN/BSRN. N indicates the number the measurements in each interval. Shadings show the range of instrumental error ($\pm$ 3 W m$^{-2}$)

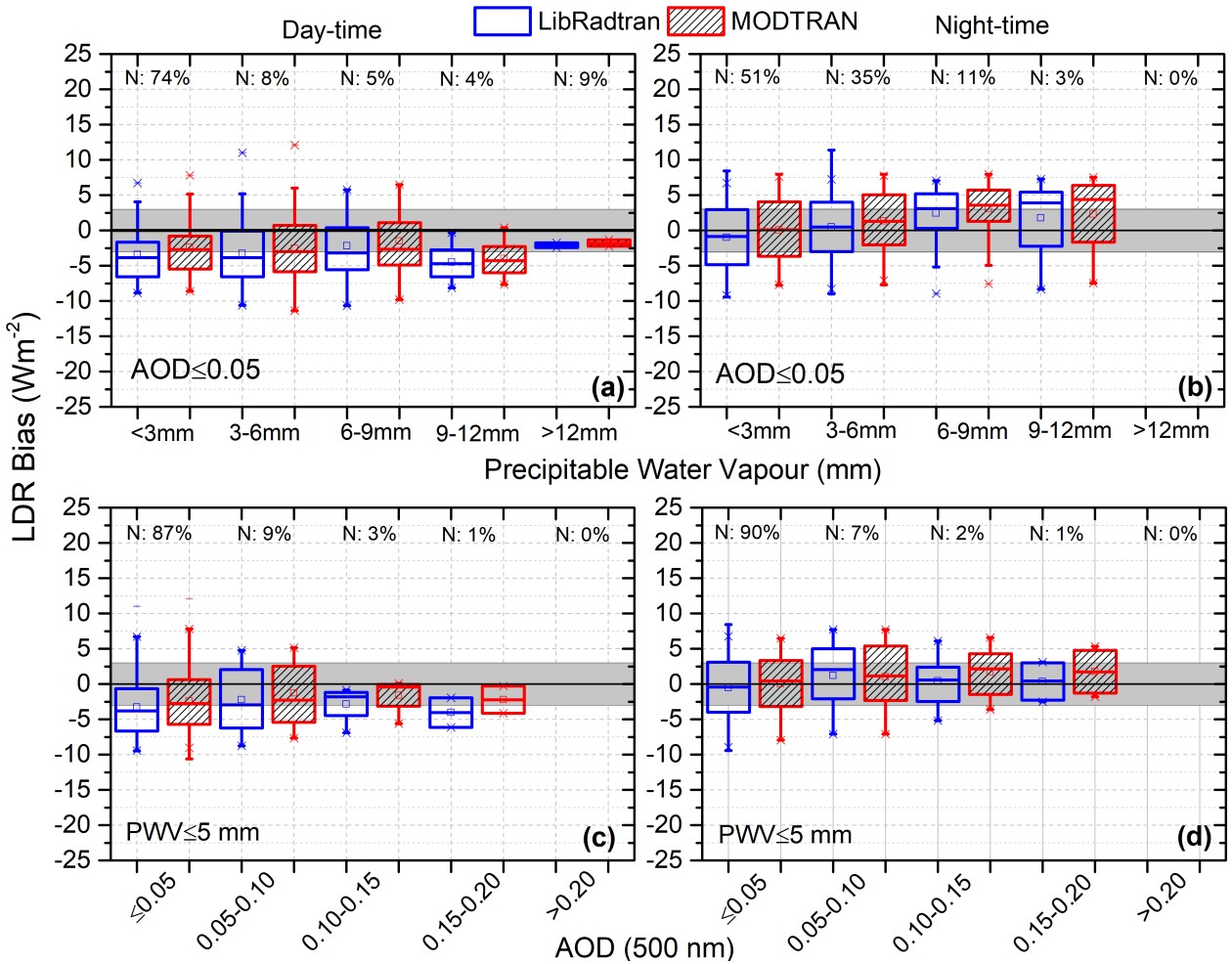

**Figure 5.** Box plot of mean LDR bias (Model-BSRN in Wm$^{-2}$) versus PWV (mm) (a) at day-time, (b) at night-time for AOD $\leq$ 0.05, and versus AOD at 500 nm (c) at day (d) night-time for PWV $\leq$ 5 nm between 2010 and 2016 at IZO. Box plots are defined as in Figure 4.

other hand, the LDR bias in function of AOD for very dry conditions (PWV $\leq$ 5 mm) according to WMO (2014b) criteria (Figures 5c and 5c).

An almost flat negative offset in LDR bias is observed in the case of AOD $\leq$ 0.05 day-time data for a relatively large range of PWV, while a larger positive bias is observed at night-time for higher PWV values (Figures 5a and 5b, respectively). These

results corroborate the dependence of LDR bias with PWV for all conditions found in Figure 4b.

The small differences in LDR bias versus PWV (close to the instrumental error) found between day-time and night-time are not currently understood. It is likely that this different behaviour between day and night may be associated with instrumental measurements (Ohmura et al., 1998; McArthur, 2005) but we do not preclude they could be also related to inaccuracies in the model input parameters during day-time, e.g., inaccuracies in the observed temperature/humidity profiles due to different

heating of the radiosonde sensors by solar radiation. Dirksen et al. (2014) studied the effects on the RS92's temperature and humidity measurements and they estimated this uncertainty to be 0.15 K for night-time temperature measurements and approximately 0.6 K at 25 km during daytime.

Concerning the LDR bias dependence with AOD for very dry conditions (PWV $\leq$ 5 mm) (Figure 5c), we observe a nearly constant negative bias at day-time, similar to that found for clean conditions (AOD $\leq$ 0.05) (Figure 5a), while the LDR bias

versus AOD at nigh-time (Figure 5d) is almost zero.

Some authors claim that dust particles might modify the transport of both shortwave and longwave radiation through the atmosphere by scattering and absorption processes and that dust radiative effects in the infrared are thus non negligible (Otto et al., 2007; Meloni et al., 2018). Our results point to an increase in the LDR bias during daytime as the AOD increases (Figure 4c) which might mean LDR underestimation by the models which would not capture the aforementioned dust absorption

and scattering processes. Notice that there is not equivalent positive trend in LDR bias for higher PWV values (Figure 4a), suggesting that this LDR bias trend is basically caused by an increase in atmospheric dust content. Unfortunately we cannot confirm these results in a dry atmosphere (removing the effect of water vapour) due to the lack of relatively high AOD data under PWV $\leq$ 5 mm conditions (Figure 5c). It would be necessary to do specific research on dust effect in LDR performing additional model simulations for different sets of dust particle size distribution and refractive index, as proposed by Meloni

et al. (2018), to confirm that the observed positive LDR bias for high AOD values during day-time is caused by mineral dust particles.

### 5.1.1 Temporal Stability

We have analyzed the temporal stability of the simulation-measurement bias time series during day-time and nigh-time in order to assess the continuity and consistency of these time series (Figure 6). We define a bias drift as the linear trend of monthly

mean bias, while the change-points (changes in the monthly mean bias time series) are analyzed by using a robust rank order change-point test (Lanzante, 1996).

By applying this change-point test, we identified October 2012 as the change point in the monthly mean bias time series at day-time and night-time for both libRadtran/BSRN and MODTRAN/BSRN time series at 99 % of confidence level (Figure 6). When analyzing the BSRN LDR and the simulated LDR times series separately, we do not observe any change in the simulated

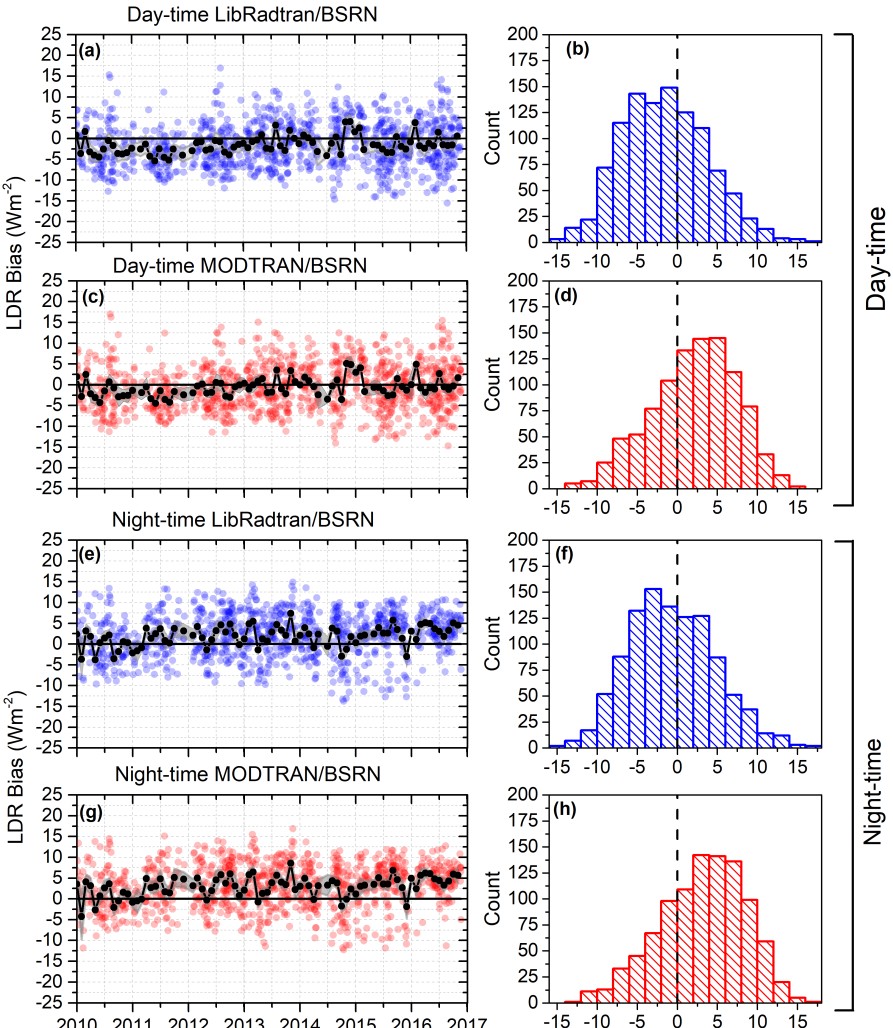

**Figure 6.** Time series and histogram of LDR bias (Model-BSRN in W m$^{-2}$) between 2010 and 2016 at IZO. The blue and red dots represent the instantaneous bias for LibRadtran/BSRN and MODTRAN/BSRN, respectively. The black dots represent the monthly mean bias. The grey shadings show the range of $\pm$ 1 SEM (standard error of the monthly mean bias).

LDR, but a change point in the BSRN LDR time series at both day-time and night-time. This change point (October 2012) coincided with a change in the location of the instrumentation within the IZO facilities. The instrument was moved to ground level during approximately a month, until the completion of the works.

## 6  Summary and Conclusions

Cloud-free longwave downward radiation (LDR) measured at the BSRN Izaña Atmospheric Observatory was compared with two complex RTMs, libRadtran v2.0.1 and MODTRAN v6, in the period 2010-2016, for a high number of cases (2062) grouped in day-time (11 UTC) dataset (in 1048 cases) and night-time (23 UTC) dataset (1014 cases). IZO is an optimal station to carry out this study, because all the model input parameters (precipitable water vapor, aerosol optical depth, total ozone, $N_2O$ in situ, $CO_2$ in situ, $CO_2$ profile and meteorological radiosondes) are measured at the station.

The agreement between measurements and simulations is excellent and very similar for both models. The mean bias (simulations-BSRN measurements) is -1.73 W m$^{-2}$ (-1.1 %) and 0.15 W m$^{-2}$ (0.1 %) for libRadtran/BSRN at day-time and night-time UTC, respectively, and -1.79 W m$^{-2}$ (-0.7 %) and 1.14 W m$^{-2}$ (0.5%) for MODTRAN/BSRN at day-time and night-time, respectively. Both comparisons showed a RMS < 3 % at day-time and < 2% at night-time. The mean bias and RMS are lower than the instrumental uncertainty ($\pm$ 3 W m$^{-2}$; 2 %; Ohmura et al. (1998); McArthur (2005)).

The MODTRAN and libRadtran performance is very similar. Both models have demonstrated to be very useful tools for LDR quality control, as well as for assessing the impact of atmospheric constituents on the Earth-atmosphere energy balance.

From our study, we state that the absolute differences between BSRN measurements and simulations depend mainly on water vapour and dust aerosols. The observed night-time difference between models and measurements of +5 W m$^{-2}$ for PWV > 10 mm supports previous measurements studies that report the existence of an offset between the World Infrared Standard Group of Pyrgeometers (WISG), which serves as reference for atmospheric longwave radiation measurements, and the SI. Concerning the possible influence of aerosols, and specifically atmospheric dust, on LDR differences between models and measurements, our preliminary results show a greater underestimation (about -3 W m$^{-2}$) of modeled LDR as AOD increases (AOD > 0.2) during day-time and dry atmosphere (PWV < 5mm), probably because the models might not correctly parametrized dust absorption and scattering processes. In fact, the LDR bias between day and night-time are currently not fully understood, and further specific analysis are needed to identify and quantify the contribution of the different possible causes.

Considering that the BSRN measurement accuracy target for LDR is $\pm$ 2 W m$^{-2}$, the average observed LDR change from 24 BSRN sites since early 1990s has been +2 W m$^{-2}$ dec$^{-1}$ (Wild, 2017) as a result of the increase of the greenhouse effect, and the CMIP5 projections estimate LDR increases between 1.7 W m$^{-2}$ dec$^{-1}$ (RCP4.5) and 2.2 W m$^{-2}$ dec$^{-1}$ (RCP8.5) over the period 2010-2013 (Wild et al., 2015; Wild, 2017), it is crucial to ensure good consistency between LDR observations and estimates with models, such as the one found in this study. Taking into account the present LDR measurement accuracy, a period of time less than two decades would be necessary for assessing completely its impact on climate change.

*Acknowledgements.* This work has been developed within the framework of the activities of the World Meteorological Organization (WMO) Commission for Instruments and Methods of Observations (CIMO) Izaña test bed for aerosols and water vapor remote sensing instruments. The authors are grateful to the libRadtran team for their assistance with the radiative transfer simulations performed in this paper. AERONET Sun photometers at Izaña have been calibrated within the AERONET Europe TNA, supported by the European Union's Horizon 2020 research and innovation program under grant agreement no. 654109 (ACTRIS-2). This research has benefited from the result of the

projects INMENSE and the project POLARMOON (funded by the Ministerio de Economía y Competividad from Spain, CGL2016-80688-P and CTM2015-66742-R, respectively) .We also acknowledge our colleague Dr. Celia Milford for improving the English language of the manuscript.

*Data availability.* The measurements of longwave downward radiation at BSRN Izaña are available at http://bsrn.awi.de

5  *Competing interests.* The authors declare that they have no conflict of interest.

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
