# Peer review of "Comparison of observed and modelled cloud-free longwave downward radiation (2010-2016) at the high mountain BSRN Izaña station"

_Geoscientific Model Development, 2017_

## Referee Comment (RC1) · Anonymous Referee #1 · 27 Feb 2018

**Comparison of observed and modelled longwave downward radiation (2010-2016) at the high mountain BSRN Izaña station**

1. General comments:

The manuscript presents a comparison of calculated and observed longwave downward radiation (LDR) for cloud-free conditions at the BSRN Izaña station using the radiative transfer models libRadtran and MODTRAN. Differences (bias ± RMSE) between calculated and observed LDR for 1014 night-time cases in the 2010-2016 period were less than 5 $Wm^{-2}$ and hence within the measurement uncertainty with the model calculations being higher compared to the observations. Discrepancies between models and observations for 1048 cloud-free day cases were slightly higher with the models yielding lower irradiances. These differences in the statistics between day and night are currently not yet understood. Finally, the models confirm the water vapor dependency of observations traceable to the World Infrared Standard Group (WISG) which has been postulated in earlier studies using windowless radiometers (e.g., Gröbner et al., 2014).

The LDR is the second largest component in the radiation budget, directly related to the greenhouse effect and hence of great importance. The accurate calculation of the cloud-free LDR is relevant in many applications. Therefore, the manuscript is significant for the community and hence appropriate for this journal. The manuscript is in general well-structured and clearly written. Graphics and tables are clear and the captions self-explanatory. Therefore, I would recommend publishing the article in GMD after minor − mainly technical - revisions.

2.) Specific comments

*i) Cloud-free detection*:

p.5: I wonder if no observational method for night time is available at this site which detects high level clouds more reliably than the Clear-Sky Index (CSI) or APCADA does. Can you comment on that? Moreover, did you really use APCADA (i.e., did you determine the diurnal and annual variability of *k* and did you calculate fractional cloud cover) or did you just use the original CSI from Marty and Philipona (2000) which only distinguishes between cloud-free and cloudy skies? Please specify. It seems to me that you used the CSI from Marty and Philipona (2000) → if this is true delete APCADA and the corresponding reference in the text.

*ii) Solar effect on the LDR and differences in the bias between the day and night comparisons of observed and calculated LDR*:

I assume that the LDR observations used in this study were shaded (according to the guidelines of BSRN), i.e. both pyrgeometers were installed on a solar tracker? It is a bit confusing because the authors state (on p.4, line 4, based on McArthur (2005)) that the CG4 filters all solar radiation and hence no shading is necessary (I see this statement anyways a bit more critical: the longwave irradiance in the direct beam of the sun is measured by any pyrgeometer and its magnitude depends on the cut-on of the filter and the solar insolation and hence on atmospheric conditions (e.g., water vapor content, cloudiness). In fact, a CG(R)4 has a higher cut-on (approximately at 4.5 µm) compared to a Eppley PIR (approximately at 4 µm) and hence the CG(R)4 measures less longwave irradiance from the sun which has been already reported in previous studies (e.g., Meloni et al., 2012). Nevertheless, a few $Wm^{-2}$ originating from the long-wave irradiance in the direct beam of the sun will be observed by a CG(R) 4 and thus it should be also operated in shaded mode). In any case, state clearly if your pyrgeometers were shaded (e.g., on p.3, line 30: '…with a shaded and ventilated broadband Kipp&Zonen…', or on page 4, line 4 after the reference of McArthur (2005)).

If the pyrgeometers were not shaded (unlikely), the long-wave irradiance in the direct beam of the sun could be a possible explanation for the small differences in the bias between the results of the day and night comparisons of measured and calculated LDR (p.11, lines 20-21/p.12, lines 1-2 and Fig. 4 or Table 4) which are in fact consistent with the results in Dürr et al, 2005. If the observations are shaded, it is reasonable that the differences between day and night are caused by additional measurement inaccuracies during daytime as stated by the authors. However, an underestimation of the models due to inaccuracies in the model input parameters during

day time (e.g., inaccuracies in the observed temperature/humidity profiles due to different heating of the radiosonde sensors by solar radiation) could be also possible (instead of instrumental inaccuracies). Could you comment on that? I would add this option at the end of the paragraph (p.12, line 2).

*Summary:*

p.3, line 30 or p.4, line 4: Specify if the LDR observations were shaded or not.

p.4, line 2: Replace 'all solar radiation' by 'most of the solar radiation'

p.11, line 13-21: I would re-arrange this paragraph and start with the night-time results first, i.e. with line 16 (in the night the uncertainties are in general smaller because of the absence of solar radiation). Then describe the results for day time.

p.11, lines 20-21/p.12, lines 1-2: If the LDR observations were not shaded the previously mentioned impact of the longwave irradiance in the solar spectrum on the LDR observations should be stated and the publication of Meloni et al. (2012) cited. If the observations are shaded, I agree with the content (but I would use '…with additional instrumental inaccuracies during day time' on p.11 line 21/p.12, line 1). In addition, I would add a sentence about possible inaccuracies in the model input parameters during day time which may result in an underestimation of the models.

3.) Technical corrections

p.1, title: add 'cloud-free' between 'modelled' and 'longwave'.

p.1, line 4 and throughout the manuscript: 'libRadtran' instead of 'LibRadtran'.

p.1, line 4: Revise sentence: 'Results show an excellent….and simulations using the radiative transfer models (RTM) libRadtran and MODTRAN V6.' (delete 'similar for both models').

p.1, line 7: '…useful tools for the quality control of LDR observations…'

p.1, line 16: cloud cover is only one aspect; I would add 'cloud type'. Furthermore, water vapor is missing.

p.2, line 2: The CG4 is nowadays termed CGR4 → use 'CG(R)4 series'

p.2, line 2: put the reference of McArthur (2005) at the end of the sentence.

p.2, lines 3-4. I would delete this sentence. The specifications for the CG(R) 4 from Kipp&Zonen may not be representative for the other types of pyrgeometers listed previously.

p.2, line 6: delete here the reference of Ohmura et al. (1998).

p.2, line 9: Reference should be Ångström, also in the reference list.

p.2, line 11, use 'e.g.,' instead of 'i.e.'

p.2, line 17, 'Stefan-Boltzmann constant'

p.2, line 27: I would put 'as model inputs' at the end of the sentence.

p.3, line 28: I would term Section 3 as 'Observational Data and Methods', then Section 3.1 'Instrument and Measurements' and Section 3.2 'Cloud-free detection'

p.3, line 4: '…with values of +1.5 and –3.2 Wm$^{-2}$ for night-time….'

p.3, line 12: rather use '…uncertainty assessment…' than '…quality assessment…'

p.3, line 13: '…temporal stability of the LDR observations…'

p.3, line 17: use 'location' instead of 'situation'.

p.3, line 21: I would use '…it has been actively contributing…'

p.3, line 21 and throughout the manuscript: an abbreviation should be define at its first occurrence in the manuscript, e.g., '…such as the Network for the Detection of Atmospheric Composition Change (NDACC; http://www.ndsc.ncep.noaa.gov/) since 1999, the Aerosol Robotic Network (AERONET, http://aeronet.gsfc.nasa.gov/) since 2004, the Total Carbon Column Observing Network (TCCON, http://www.tccon.caltech.edu/) since 2007,…'. Later, just use the abbreviation.

p.3, line 26: Revise reference (also in the reference list). Should be read 'WMO' or 'CIMO', I guess.

p.4, line 6: '… at the Physikalisch-Meteorologisches Observatorium Davos/World Radiation Center (PMOD/WRC).'

p.4, line 9: The reference is from 2002, I guess. Revise also in the reference list.

p.5, line 1: Here, I would use only the reference of Dürr and Philipona (2004) but only if you have really used APCACA (see my previous comments). Insert the reference of Marty and Philipona in line 4 (after 'at the station'). If you have used the CSI from Marty and Philipona (2000) replace APCADA and the corresponding reference with 'Clear- Sky Index (CSI) (Marty and Philipona (2000)' in line 1, p.5.

p.5, line 10: 'Stefan-Boltzmann'

p.5, line 10: $\varepsilon_{AD}$ is an altitude-dependent emittance of a completely dry atmosphere ($\varepsilon_{AC}$ is the apparent emittance of a cloud-free sky)

p.5, lines 11/12: Revise this sentence, e.g.: 'A CSI Index $\leq 1$ and $> 1$ indicates cloud-free and cloud-sky, respectively.'

p.5, line 16: '… consists of…'

p.5, line 28: delete 'models'.

p.6, line 7: Hasn't the band model used in MODTRAN 6 a resolution of $0.1 cm^{-1}$?

p.7, line 10: The site of the radiosonde launch is located at sea level, more than 2000 m lower the IZO. I assume that you cut the profiles at the altitude of IZO to assimilate the profiles into the RTM?

p. 7, line 18: NDACC has been already defined on p. 3.

p.7, line 31: delete 'one'.

p.8, line 15: delete 'the'.

p.10, line 5: Did you average the observations over a certain time period (e.g., 30 minutes) in order to validate the RTM calculations? Or did you use the 1 min observations? Specify.

p.10, line 7: you may better use '… and R of 0.999, and are more consistent during nighttime'.

p.11, Table 4 (Caption): The number of day-time calculations given here (1075 cases) is not consistent with those given in the abstract, Section 5 (1048, p.10) and Section 6.

p.12, line 11: Could you specify what was changed in the location of the instrumentation in 2012?

p.14, line 3: 'supports'.

p.14, line 4: I would add 'However, the differences between day and night are currently not yet understood.'

p.17, line 37: Specify journal/meeting event of publication/presentation of Redondas and Cede.

References:

Dürr, B. and Philipona, R.: Automatic cloud amount detection by surface longwave downward radiation measurements, Journal of Geophysical Research: Atmospheres, 109, https://doi.org/10.1029/2003JD004182, 2004.

Dürr, B., Philipona, R., Schubiger, F., and 5 Ohmura, A.: Comparison of modeled and observed cloud-free longwave downward radiation over the Alps, Meteorologische Zeitschrift, 14, 47–55, https://doi.org/https://doi.org/10.1127/0941-2948/2005/0014-0047, 2005.

Gröbner, J., Reda, I., Wacker, S., Nyeki, S., Behrens, K., and Gorman, J.: A new absolute reference for atmospheric longwave irradiance measurements with traceability to SI units, Journal of Geophysical Research: Atmospheres, **119**, 7083–7090, 2014.

Marty, C. and Philipona, R.: Clear-sky index to separate clear-sky from cloudy-sky situations in climate research, Geophysical Research Letters, 27, 2649–2652, 2000.

McArthur, L.: Baseline Surface Radiation Network (BSRN) Operations Manual V2.1,World Climate Research Programme,wmo/td-no. 1274 25 ed., WCRP-121, 2005.

Meloni, D., di Biaggio, C., di Sarra, A., Monteleone, F., Pace, G., Sferlazzo, D.M., Accounting for the Solar Radiation Influence on Downward Longwave Irradiance Measurements by Pyrgeometers, J. Atmos. Oceanic Technol. **29**, 2012, doi: http://dx.doi. org/10.1175/JTECH-d-11-00216.1.

---

## Short Comment (SC1) · 15 Mar 2018

GMD does not necessary require for a Model evaluation paper to make statements about code availability. This is applied under the assumption that the manuscript is referencing a paper that describes the model being evaluated and that this paper states how to obtain access to the program code of the model. As this does not apply for this manuscript, the authors need to explain how to access the code. As stated in https://www.geoscientific-model-development.net/about/manuscript_types.html for "Model description papers" the preferred option is that authors upload their code and the data as supplement.

[Figure]

Lutz Gross GMD Executive Editor

---

## Referee Comment (RC2) · Anonymous Referee #2 · 16 Mar 2018

GMD-2017-303 review The manuscript presents a concise comparison of 7 years of downward longwave radiation measurements obtained at the Izania Atmospheric Observatory to two high resolution radiative models using other measured parameters at the site. The results show agreement between the two models and pyrgeometer measurements to within their demonstrated uncertainties. This manuscript only needs minor adjustments for publication and will be of great benefit to both the modelling and measurement communities.

General comments 1. There is no indication of what DLR measureands are used in the comparison. Are they single sample, minute averages or longer averages. There

is reference to 'instantaneous' measurements but such measurements do not exist as most data acquisition systems integrate over a small but finite period. For example, there is reference to 1-minute surface measurements in 3.0.1 but are they averages or single samples. 2. On occasions 'accuracies' are given a quantitative value. In ISO accuracies are a qualitative (good, bad, indifferent) not quantitative. Just because a manufacturer incorrectly uses accuracy as a quantitative term is no reason to repeat bad practise. 3. 'Temporal resolution' and 'temporal frequency' are used in 4.1 lines 7 to 15 - but what one thinks is meant is sampling rate. 4. While the AOD at 500 nm is used there is no indication of the aerosol model (i.e. distribution) that scales in the IR. 5. Figure 3 shows a standard X vs Y plot of various comparison parameters. It would be more instructive as (Y-X) vs X plots with a (Y-X) = zero line. 6. Table 4. Unless one of the variables is the 'truth' then the RSME are really root mean square differences. 7. 5.1.1 - while the step jump on relocation was detected there does not appear to be any comment on the different pygeometers. Was one replaced with another? If not, see point 1 above as it is not clear what measurements were used; a mean between the two?? If one was replaced with another then it would be worth saying that no jump in differences were detected when replacing an instrument. 8. Section 6 line 1-5: 'suggest a scale change of the WISG' - this is an erroneous statement as the WISG is an interim scale until a better one can be found. It might be better to rephrase it to 'The. . . . . . support previous measurement studies that suggest an offset of the WISG to the SI.'

Specific suggestions.

a. Abstract line 5: delete 'similar'. b. Abstract last sentence: move 'for precipitable water vapor (PWV) >10 mm,' to the start of the sentence. c. All references citing 'World infrared standard group' should be replaced with 'World Infrared Standard Group' or after the first use WISG. d. Page 11 line 18: the ; before Nyeki et al should be replaced with 'and'. e. There are a number of other typos that one hopes and editor can correct.

---

## Author Comment (AC1) · 26 Mar 2018

**Anonymous Referee #1:**

**GENERAL COMMENTS**

The manuscript presents a comparison of calculated and observed longwave downward radiation (LDR) for cloud-free conditions at the BSRN Izaña station using the radiative transfer models libRadtran and MODTRAN. Differences(bias ±RMSE)between calculated and observed LDR for 1014 night-time cases in the 2010-2016 period were less than5 Wm-2and hence within the measurement uncertainty with the model calculations being higher compared to the observations. Discrepancies between models and observationsfor1048 cloud-free day cases were slightly higher with the models yielding lower irradiances. These differences in the statistics between day and night are currently not yet understood. Finally, the models confirm the water vapour dependency of observations traceable to the World Infrared Standard Group (WISG) which has been postulated in earlier studies using windowless radiometers (e.g., Gröbner et al., 2014).

The LDR is the second largest component in the radiation budget, directly related to the greenhouse effect and hence of great importance. The accurate calculation of the cloud-free LDR is relevant in many applications. Therefore, the manuscript is significant for the community and hence appropriate for this journal. The manuscripts in general well-structured and clearly written. Graphics and tables are clear and the captions self-explanatory. Therefore, I would recommend publishing the article in GMD after minor–mainly technical -revisions.

*Authors:  We appreciate the positive and constructive comments of the Referee.  Here we discuss and respond to his/her specific comments and technical corrections.*

 2.) SPECIFIC COMMENTS

i) Cloud-free detection:

p.5: I wonder if no observational method for night time is available at this site which detects high level clouds more reliably than the Clear-Sky Index (CSI) or APCADA does. Can you comment on that? Moreover, did you really use APCADA (i.e., did you determine the diurnal and annual variability of k and did you calculate fractional cloud cover) or did you just use the original CSI from Marty and Philipona (2000) which only distinguishes between cloud-free and

**cloudy skies? Please specify. It seems to me that you used the CSI from Marty and Philipona (2000) if this is true delete APCADA and the corresponding reference in the text.**

*Authors:  A SONA total-sky camera (Automatic Cloud Observation System; González et al., 2013) which takes an image every 5 minutes during all day, has been operating at Izaña Observatory during the time period of this study. The SONA's images have been used to check the cloud-free results obtained with the Clear-Sky Index following the Long and Ackerman method. We have reviewed the recorded SONA images by visual examination during the day-time (11 UTC) and night-time (23 UTC), because the camera is sensible enough to observe clouds during night period. We will incorporate this information into the final manuscript as shown hereinafter.*

*The referee is right and the method used to detect cloud-free and cloud-skies was the one developed by Marty and Philipona (2000) and not the APCADA (Dürr and Philipona, 2004).*

*For this reason, the section "Cloud-free detection" has been modified as follows:*

*"The cloud-free days were detected by using the algorithm developed by Marty and Philipona (2000). A Clear-Sky Index (CSI) is calculated to separate cloud-free days from cloudy days using accurate measurements of LDR in conjunction with air temperature and relative humidity values measured at the station. The CSI index is defined as:*

$$CSI = \frac{\varepsilon_A}{\varepsilon_{AC}} \qquad\qquad (2)$$

*where*

$$\varepsilon_A = \frac{LDR}{\sigma_b T^4} \qquad\qquad (3)$$

$$\varepsilon_{AC} = \varepsilon_{AD} + k(e/T)^{1/8} \qquad\qquad (4)$$

*where $\sigma_b$ is the Stefan-Boltzmann constant, $T$ is the air temperature (K), $\varepsilon_{AD}$ is an altitude-dependent emittance of a completely dry atmosphere, $e$ is water vapor pressure (Pa) and $k$ is a constant coefficient dependent on the location. If CSI Index ≤ 1 indicates cloud-free (no clouds) and if CSI Index >1 indicates cloud-sky (overcast) (Marty and Philipona, 2000).*

*In order to calculate $\varepsilon_{AC}$, this method requires the evaluation of $\varepsilon_{AD}$ and $k$, as shown in equation (4). A sample of known cloud-free days is used to plot $\varepsilon_{AC}$ against $e/T$ (Figure 2). The cloud-free condition of this sample is assured by applying the Long and Ackerman's method (Long and Ackerman, 2000; adapted for IZO by García et al. (2014)). This method is based on surface measurements of global and diffuse solar radiation with a 1-min sampling period and consists of in four individual tests applied to normalized global radiation magnitude, maximum diffuse radiation, change in global radiation with time, and normalized diffuse radiation ratio variability. We have*

*considered the period 2010-2016 at 11 UTC to determine the fitting coefficients of equation (4) obtaining the following relationship (Figure 2):*

$$\varepsilon_{AC} = 0.218 + 0.385(e/T)^{1/8} \qquad\qquad (5)$$

*Despite the $\varepsilon_{AD}$ depends on the altitude of the station we have obtained for IZO a value of 0.218, similar to the values obtained by Marty and Philipona (2000) for stations located between 2230 and 2540 m (0.22 and 0.211, respectively).*

*Once we have adjusted the coefficients, the cloud-free cases were selected with a combination of Long and Ackerman and CSI methods. At day-time, we have used the Long and Ackerman one, taking into account for each day the period 11-13 UTC. At night-time the CSI was applied in the period 23-01 UTC. These results have been checked by visual examination of 5-minute total sky images obtained with a SONA camera (Automatic Cloud Observation System; González et al., 2013) installed at IZO. We found that both methods detect 97% of the visually selected cases. A total of 1161 and 1083 cases were detected for day-time and night-time, respectively, in the period 2010-2016 "*

**ii) Solar effect on the LDR and differences in the bias between the day and night comparisons of observed and calculated LDR:**

**I assume that the LDR observations used in this study were shaded (according to the guidelines of BSRN), i.e. both pyrgeometers were installed on a solar tracker? It is a bit confusing because the authors state (on p.4,line4, based on McArthur (2005)) that the CG4 filters all solar radiation and hence no shading is necessary (I see this statement anyways a bit more critical: the longwave irradiance in the direct beam of the sun is measured by any pyrgeometer and its magnitude depends on the cut-on of the filter and the solar insolation and hence on atmospheric conditions (e.g., water vapor content, cloudiness).In fact, a CG(R)4 has a higher cut-on (approximately at 4.5µm) compared to a Eppley PIR (approximately at 4µm)and hence the CG(R)4 measures less longwave irradiance from the sun which has been already reported in previous studies (e.g., Meloni et al., 2012). Nevertheless, a few Wm-2 originating from the long-wave irradiance in the direct beam of the sun will be observed by a CG(R) 4 and thus it should be also operated in shaded mode).In any case, state clearly if your pyrgeometers were shaded (e.g., on p.3, line 30: '…with a shaded and ventilated broadband Kipp & Zonen…', or on page 4, line 4 after the reference of McArthur (2005)).**

**If the pyrgeometers were not shaded (unlikely), the long-wave irradiance in the direct beam of the sun could be a possible explanation for the small differences in the bias between the results of the day and night comparisons of measured and calculated LDR (p.11, lines20-21/p.12, lines 1-2and Fig. 4 or Table 4) which are in fact consistent with the results in Dürr et al, 2005. If the observations are shaded, it is reasonable that the differences between day and night are caused by additional measurement inaccuracies during daytime as stated by the authors. However, an underestimation of the models due to inaccuracies in the model input parameters during day time (e.g., inaccuracies in the observed temperature/humidity profiles due to different heating of the radiosonde sensors by solar radiation) could be also possible**

**(instead of instrumental inaccuracies). Could you comment on that? I would add this option at the end of the paragraph (p.12, line 2).**

*Authors: The authors appreciate these interesting comments and remarks.*

*The LDR observations have been performed with CGR4 pyrgeometers installed on a solar shaded tracker. This information has been added to the final manuscript in Pag. 3 Line 30 as follows:*

> *"The LDR measurements used in this study have been performed at the Izaña BSRN (#61, IZA; http://www.bsrn.aemet.es) (García et al., 2012) with a ventilated broadband Kipp & Zonen CG4 pyrgeometer (onwards, CGR4) mounted on a sun tracker equipped with dome shading. This instrument uses a specially designed silicon window which provides a 180° field of view (although not hemispherical) with good cosine response. A diamond-like surface protects the outer surface of the window, while the inner surface filters most of solar radiation. The design of the instrument is such that solar radiation absorbed by the windows is conducted away to reduce the solar heating effect. This fact reduces the need for dome heating correction terms and shading from the sun (McArthur, 2005)."*

*Pag 12 Line 2:*

> *"The small differences observed in the evolution of the bias with the PWV (close to the instrumental error) found between day-time and night-time may be associated with instrumental measurements (Ohmura et al., 1998; McArthur, 2005) and we do not preclude they could be also related to inaccuracies in the model input parameters during day-time, e.g., inaccuracies in the observed temperature/humidity profiles due to different heating of the radiosonde sensors by solar radiation. Dirksen et al. (2014) studied the effects on the RS92's temperature and humidity measurements and they estimated this uncertainty to be 0.15 K for night-time temperature measurements and approximately 0.6 K at 25 km during daytime. "*

*Pag 14 Line 4:*

> *"The differences between day and night-time are currently not understood. Further specific analysis are needed to identify and quantify the contribution of the different possible causes for the observed differences which are outside the scope of this work."*

**SUMMARY:**

**p.3, line 30 or p.4, line 4: Specify if the LDR observations were shaded or not.**

*Authors: See previous answer (SPECIFIC COMMENTS ii).*

**p.4, line 2: Replace 'all solar radiation' by 'most of the solar radiation'**
*Authors: Done.*

**p.11, line 13-21:** I would re-arrange this paragraph and start with the night-time results first, i.e. with line 16 (in the night the uncertainties are in general smaller because of the absence of solar radiation). Then describe the results for day time.

*Authors: The authors think it is not convenient to discuss night-time results before day-time results, for consistency with the rest of the manuscript.*

**p.11,lines20-21/p.12, lines 1-2:** If the LDR observations were not shaded the previously mentioned impact of the longwave irradiance in the solar spectrum on the LDR observations should be stated and the publication of Meloni et al. (2012) cited. If the observations are shaded, I agree with the content (but I would use '…with additional instrumental inaccuracies during daytime' on p.11 line 21/p.12, line 1). In addition, I would add a sentence about possible inaccuracies in the model input parameters during day time which may result in an underestimation of the models.

*Authors: See previous answer (SPECIFIC COMMENTS ii).*

**3.) TECHNICAL CORRECTIONS**

**p.1, title:** add 'cloud-free' between 'modelled' and 'longwave'.
*Authors: Done.*

**p.1, line 4 and throughout the manuscript:** 'libRadtran' instead of 'LibRadtran'.
*Authors: Done.*

**p.1, line 4:** Revise sentence: 'Results show an excellent….and simulations using the radiative transfer models (RTM) libRadtran and MODTRAN V6.'( delete 'similar for both models').
*Authors: Done.*

**p.1, line 7:** '…useful tools for the quality control of LDR observations…'
*Authors: Done.*

**p.1, line 16:** cloud cover is only one aspect; I would add 'cloud type'. Furthermore, water vapor is missing.

*Authors: We fully agree. Following the Referee's recommendation, the authors have added the underlined information into the following paragraph:*

> *"The longwave downward radiation (LDR) at the Earth's surface is a key component in land-atmosphere interaction processes, and is crucial in the surface energy budget and global climate change, because the changes in the LDR values may be related to changes in cloud-cover, cloud type, water vapour, temperature, and the increase of anthropogenic greenhouse gas concentrations in the atmosphere (Wild et al., 1997; Marty et al., 2003)."*

**p.2, line 2:** The CG4 is nowadays termed CGR4 use 'CG(R)4 series'
*Authors: Done.*

**p.2, line 2:** put the reference of McArthur (2005) at the end of the sentence.

*Authors: Done.*

**p.2, lines 3-4. I would delete this sentence. The specifications for the CG(R)4 from Kipp & Zonen may not be representative for the other types of pyrgeometers listed previously.**
*Authors: Done.*

**p.2, line 6: delete here the reference of Ohmura et al. (1998).**
*Authors: Done.*

**p.2, line 9: Reference should be Ångström, also in the reference list.**
*Authors: Done.*

**p.2, line 11, use 'e.g.,' instead of 'i.e.'**
*Authors: Done.*

**p.2, line 17, 'Stefan-Boltzmann constant'**
*Authors: Done.*

**p.2, line 27: I would put 'as model inputs' at the end of the sentence.**
*Authors: Done.*

**p.3, line 28: I would term Section 3 as 'Observational Data and Methods', then Section 3.1 'Instrument and Measurements' and Section 3.2'Cloud-free detection'**
*Authors: Done.*

**p.3, line 4: '…with values of +1.5and –3.2Wm-2for night-time….'**
*Authors: Done.*

**p.3, line 12: rather use'…uncertainty assessment…'than '…quality assessment…'**
*Authors: Done.*

**p.3, line 13: '…temporal stability of the LDR observations…'**
*Authors: Done.*

**p.3, line 17: use 'location' instead of 'situation'.**
*Authors: Done.*

**p.3, line21: I would use '…it has been actively contributing…'**
*Authors: Done.*

**p.3, line 21 and throughout the manuscript: an abbreviation should be define at its first occurrence in the manuscript, e.g.,'…such as the Network for the Detection of Atmospheric Composition Change(NDACC; http://www.ndsc.ncep.noaa.gov/)since 1999,the Aerosol Robotic Network(AERONET, http://aeronet.gsfc.nasa.gov/) since 2004, the Total Carbon Column Observing Network (TCCON, http://www.tccon.caltech.edu/)since2007,…'. Later, just use the abbreviation.**
*Authors: Done.*

**p.3, line 26: Revise reference (also in the reference list). Should be read 'WMO' or 'CIMO', I guess.**
*Authors: Done.*

**p.4, line 6: '…at the Physikalisch-Meteorologisches Observatorium Davos/World Radiation Center (PMOD/WRC).'**

*Authors: Done.*

**p.4, line 9: The reference is from 2002, I guess. Revise also in the reference list.**

*Authors: Done.*

**p.5, line 1: Here, I would use only the reference of Dürr and Philipona (2004) but only if you have really used APCACA (see my previous comments). Insert the reference of Marty and Philipona inline 4 (after 'at the station').If you have used the CSI from Marty and Philipona (2000) replace APCADA and the corresponding reference with 'Clear-Sky Index (CSI) (Marty and Philipona(2000)'in line 1, p.5.**

*Authors: Done. See previous answer (SPECIFIC COMMENTS i).*

**p.5, line 10: 'Stefan-Boltzmann'**

*Authors: Done.*

**p.5, line 10: $\varepsilon_{AD}$ is an altitude-dependent emittance of a completely dry atmosphere ($\varepsilon_{AC}$ is the apparent emittance of a cloud-free sky)**

*Authors: Done. See previous answer (SPECIFIC COMMENTS i).*

**p.5, lines11/12: Revise this sentence, e.g.: 'A CSI Index ≤1 and >1indicates cloud-free and cloud-sky, respectively.'**

*Authors: Done. See previous answer (SPECIFIC COMMENTS i).*

**p.5, line 16: '…consists of…'**

*Authors: Done.*

**p.5, line 28: delete 'models'.**

*Authors: Done.*

**p.6, line 7: Hasn't the band model used in MODTRAN 6 a resolution of 0.1cm-1?**

*Authors:  Yes, MODTRAN 6 has resolution of 0.1 $cm^{-1}$, but we use a resolution of 1 $cm^{-1}$ in order to reduce de computational time, taking into account that the integrated LDR using 1 $cm^{-1}$ resolution differs not significantly from the 0.1 $cm^{-1}$ one.*

**p.7, line 10: The site of the radiosonde launch is located at sea level, more than 2000 m lower the IZO. I assume that you cut the profiles at the altitude of IZO to assimilate the profiles into the RTM?**

*Authors:  Yes. The radiosonde profiles have been used from the altitude of IZO (2373 m a.s.l.).  The authors have been added the following sentence:*

> *"In this work, we have used the AEMET's meteorological radiosondes dataset. Radiosondes are launched twice a day, at 11 and 23 UTC at the Güimar station (WMO GRUAN station #60018, 105 m a.s.l.). This station is located at the coastline, approximately 15 km to the southeast of IZO. Vertical profiles of pressure, temperature and relative humidity were measured using Vaisala RS92 radiosondes (Cuevas et al., 2015; Carrillo et al., 2016). We have used the radiosonde profiles from the altitude of IZO (2373 m .a.s.l.)"*

**p. 7, line 18: NDACC has been already defined on p. 3.**

*Authors: Done.*

**p.7, line 31: delete 'one'.**
*Authors: Done.*

**p.8, line 15: delete 'the'.**
*Authors: Done.*

**p.10, line 5: Did you average the observations over a certain time period (e.g., 30 minutes) in order to validate the RTM calculations? Or did you use the 1 min observations? Specify.**

*Authors: Yes. We have averaged in a 30 minutes period the LDR observations from 11:00 to 11:30 and from 23:00 to 23:30 UTC to match the flight time of the radiosonde over IZO. This information has been added to the final manuscript as follows:*

> *"In this section, we present the comparison between LDR measured with BSRN and simulated with LibRadtran and MODTRAN, considering the available and coincident cloud-free BSRN at day-time and night-time, and the inputs indicated in section 4.1 at IZO between 2010 and 2016. A total of 1048 measurements at day-time, and 1014 measurements at night-time were used. The observations were averaged in a time period of 30 minutes, in order match the flight time of the radiosonde over IZO. In particular, we averaged over 11:00-11:30 UTC and 23:00-23:30 UTC periods, for day-time and night-time measurements, respectively. .."*

**p.10, line 7: you may better use '…and R of 0.999, and are more consistent during nighttime'.**

*Authors: Done*

**p.11, Table 4 (Caption): The number of day-time calculations given here (1075 cases) is not consistent with those given in the abstract, Section 5 (1048, p.10) and Section 6.**

*Authors: Done*

**p.12, line 11: Could you specify what was changed in the location of the instrumentation in 2012?**
*Authors: Due to improvement works at IZO the instrumentation was moved for a short period of time (approximately a month) from the tower to a nearby platform at ground level. Once the works ended the instrumentation was reinstalled in its original location. This information has been added to the final manuscript as follows:*

> *"… When analyzing the BSRN LDR and the simulated LDR data time series separately, we do not observe any change in the simulated LDR, but a change point in the BSRN LDR data time series at both day-time and night-time. This change point (October 2012) coincided with a change in the location of the instrumentation within the IZO facilities. The instrument was moved to ground level during approximately a month, until the works ended…."*

**p.14, line 3: 'supports'.**

*Authors: Done*

**p.14, line 4: I would add 'However, the differences between day and night are currently not yet understood.'**

*Authors: The authors have been added this sentence to the final manuscript (See previous answer (SPECIFIC COMMENTS ii).*

**p.17, line 37: Specify journal/meeting event of publication/presentation of Redondas and Cede.**

*Authors: Done*

---

## Author Comment (AC2) · 26 Mar 2018

**Dr. L. Gross**

**GMD does not necessary require for a Model evaluation paper to make statements about code availability. This is applied under the assumption that the manuscript is referencing a paper that describes the model being evaluated and that this paper states how to obtain access to the program code of the model. As this does not apply for this manuscript, the authors need to explain how to access the code. As stated in https://www.geoscientific-model-development.net/about/manuscript_types.html for "Model description papers" the preferred option is that authors upload their code and the data as supplement.**

*Authors: The authors take into consideration the editor's request.*

*However, in this work we present the comparison between measured and simulated longwave downward radiation using two radiative transfer models: LibRadtran and MODTRAN.*

*The LibRadtran model is freely available on the web: http://www.libradtran.org; Mayer and Kylling (2005). This information has been included in the manuscript Page 6 Line 1-7.*

*The MODTRAN model is only available under the commercial agreement with Spectral Sciences, Inc. (http://modtran.spectral.com; Berk et al., 2000, 2008, 2013, 2015; Berk and Hawes, 2017).*

*Since these two models have not been developed by the authors of the paper, it is not possible to add their code.*

*These models have been extensively described in the following references which were included in the paper.*

*References:*

1. *Berk, A., Acharya, P. K., Anderson, G., Chetwynd, J. H., and Hoke, M. L.: Reformulation of the MODTRAN band model for higher spectral resolution, in: Proceedings spue the international society for optical engineering, pp. 190–198, International Society for Optical Engineering; 1999, 2000.*
2. *Berk, A., P.K. Acharya, L.S. Bernstein, G.P. Anderson, P. Lewis, J.H. Chetwynd, and M.L. Hoke, "Band Model Method for Modeling Atmospheric Propagation at Arbitrarily Fine Spectral Resolution," U.S. Patent #7433806, issued October 7, 2008.*

3. *Berk, P. Conforti, R. Kennett, T. Perkins, F. Hawes, and J. van den Bosch,* "MODTRAN6: a major upgrade of the MODTRAN radiative transfer code," *Proc. SPIE 9088, Algorithms and Technologies for Multispectral, Hyperspectral, and Ultraspectral Imagery XX, 90880H (June 13, 2014); doi:10.1117/12.2050433.*

4. *Alexander Berk, Patrick Conforti, and Fred Hawes,* "An accelerated line-by-line option for MODTRAN combining on-the-fly generation of line center absorption with 0.1 cm-1 bins and pre-computed line tails," *Proc. SPIE 9471, Algorithms and Technologies for Multispectral, Hyperspectral, and Ultraspectral Imagery XXI, 947217 (May 21, 2015); doi:10.1117/12.2177444*

5. *Berk, A. and Hawes, F.: Validation of MODTRAN® 6 and its line-by-line algorithm, Journal of Quantitative Spectroscopy and Radiative Transfer, 203, 542–556, 2017.*

6. *Mayer, B. and Kylling, A.: Technical note: The libRadtran software package for radiative transfer calculations – description and examples of use, Atmospheric Chemistry and Physics, 5, 1855–1877, https://doi.org/10.5194/acp-5-1855-2005, http://www.atmos-chem-phys.net/5/1855/2005/, 2005.*

---

## Author Comment (AC3) · 26 Mar 2018

**Anonymous Referee #2:**

**GMD-2017-303 review**

**The manuscript presents a concise comparison of 7 years of downward longwave radiation measurements obtained at the Izaña Atmospheric Observatory to two high resolution radiative models using other measured parameters at the site. The results show agreement between the two models and pyrgeometer measurements to within their demonstrated uncertainties. This manuscript only needs minor adjustments for publication and will be of great benefit to both the modelling and measurement communities.**

*Authors: The authors acknowledge the referee's constructive comments, and in the followings, we discuss and respond to the general comments and specific suggestions.*

**GENERAL COMENTS**

**1. There is no indication of what DLR measureands are used in the comparison. Are they single sample, minute averages or longer averages. There is reference to 'instantaneous' measurements but such measurements do not exist as most data acquisition systems integrate over a small but finite period. For example, there is reference to 1-minute surface measurements in 3.0.1 but are they averages or single samples.**

*Authors: The authors have averaged in a 30 minutes period the LDR observations from 11:00 to 11:30 and 23:00 to 23:30 UTC to match the flight time of the radiosonde over IZO. This information has been added to the final manuscript as follows:*

> *"In this section, we present the comparison between LDR measured with BSRN and simulated with LibRadtran and MODTRAN, considering the available and coincident cloud-free BSRN at day-time and night-time, and the inputs indicated in section 4.1 at IZO between 2010 and 2016. A total of 1048 measurements at day-time, and 1014 measurements at night-time were used. The observations were averaged in a time period of 30 minutes, in order match the flight time of the radiosonde over IZO. In particular, we averaged over 11:00-11:30 UTC and 23:00-23:30 UTC periods, for day-time and night-time measurements, respectively. .."*

**2. On occasions 'accuracies' are given a quantitative value. In ISO accuracies are a qualitative (good, bad, indifferent) not quantitative. Just because a manufacturer incorrectly uses accuracy as a quantitative term is no reason to repeat bad practise.**

*Authors: The authors agree with this comment. We have replaced the following uses of accuracy:*

> *Page 2, Line 4:*
>
> *"…The spectral range covers from 4 to 42 μm with an expected sensitivity of 5 to 15 μV/Wm$^{-2}$, an uncertainty < 3% for daily totals, and uncertainty < 7.5 Wm$^{-2}$"*
>
> *Page 2, Line 18:*
>
> *"…The above mentioned parameterizations show uncertainties ranging from 9% to 15 % in low altitude sites…"*

**3. 'Temporal resolution' and 'temporal frequency' are used in 4.1 lines 7 to 15 - but what one thinks is meant is sampling rate.**

*Authors: We agree, and have modified the two sentences as follows:*

> *Line 7:*
>
> *"In this work, we have used the AEMET's meteorological radiosondes dataset. Radiosondes are launched twice a day, at 11 and 23 UTC at the Güimar station…"*
>
> *Line 15:*
>
> *Since January 2009, the PWV has been obtained every 1h at IZO from a GNSS (GPS-GLONASS) receiver considering GPS precise orbits (Romero Campos et al., 2009)*

**4. While the AOD at 500 nm is used there is no indication of the aerosol model (i.e. distribution) that scales in the IR.**

*Authors: Following the referee's recommendation, we have added this information in the final manuscript as follows:*

> *"Atmospheric aerosols are included in the simulation process by means of the column-integrated AOD extracted from AERONET (Level 2.0 of version 2, cloud screened and quality ensured). The AOD is obtained from solar observations performed with CIMEL sunphotometers at different wavelengths (Holben et al., 1998; Dubovik and King, 2000; Dubovik et al., 2006). The Shetle's aerosol model (Shetle, 1989) has been used in this study. The default properties are: rural type aerosol in the boundary layer, background aerosol above 2 km, spring-summer conditions and a visibility of 50 km. In this work, AOD at 500 nm has been used as model input. For day-time we have used the nearest AOD value to 11 UTC, and for night-time the last available AOD value of the day."*

*Shettle, E.: Models of aerosols, clouds and precipitation for atmospheric propagation studies, in: Atmospheric propagation in the uv, visible, ir and mm-region and related system aspects, no. 454 in AGARD Conference Proceedings, 1989.*

**5. Figure 3 shows a standard X vs Y plot of various comparison parameters. It would be more instructive as (Y-X) vs X plots with a (Y-X) = zero line.**

*Authors:   Following the referee's recommendation, we have done a new Figure 3. However, the authors think that as the information of this new Figure is already provided by Figure 5 of the original manuscript, it is more convenient to keep Figure 3 as it was presented in the original manuscript.*

[Figure]

*Figure.-  Difference between LDR ($Wm^{-2}$) simulations with libRadtran (blue color) and LDR BSRN ($Wm^{-2}$) at cloud-free (a) day-time and (d) night-time. Difference between MODTRAN LDR ($Wm^{-2}$) (red color) and BSRN LDR ($Wm^{-2}$) at (b) day-time and (e) night-time. Difference between MODTRAN LDR ($Wm^{-2}$) and libRadtran LDR (($Wm^{-2}$) (black color) at (c) day-time and (f) night-time*

**6. Table 4. Unless one of the variables is the 'truth' then the RSME are really root mean square differences.**

*Authors:  Thank you for the comment. The authors have been changed the root mean square error (RMSE) by root mean square of the bias (RMS) in the final manuscript.*

**7. 5.1.1 - while the step jump on relocation was detected there does not appear to be any comment on the different pygeometers. Was one replaced with another? If not, see point 1 above as it is not clear what measurements were used; a mean between the two?? If one was replaced with another then it would be worth saying that no jump in differences were detected when replacing an instrument.**

*Authors:  The pygeometer was not replaced by any other one. The BSRN instrumentation was moved from the tower to a platform at ground level, because maintenance works at the tower, for a short period of time (approximately one month). Once the works ended the instrumentation was moved back to its original location. The BSRN operations were not interrupted during this period of time.*

> *"… When analyzing the BSRN LDR and the simulated LDR data time series separately, we do not observe any change in the simulated LDR, but a change point in the BSRN LDR time series at both day-time and night-time. This change point (October 2012) coincided with a change in the location of the instrumentation within the IZO facilities. The instrument was moved to ground level during approximately a month, until the completion of the works…."*

**8. Section 6 line 1-5: 'suggest a scale change of the WISG' - this is an erroneous statement as the WISG is an interim scale until a better one can be found. It might be better to rephrase it to 'The support previous measurement studies that suggest an offset of the WISG to the SI.'**

*Authors:  Following the referee's suggestion, we have modified this information in the final manuscript as follows:*

> *"..The observed night-time difference between models and measurements of +5 $Wm^{-2}$ for PWV> 10 mm supports previous measurements studies that report the existence of an offset between the World Infrared Standard Group of Pyrgeometers (WISG), which serves as reference for atmospheric longwave radiation measurements, and the SI."*

**SPECIFIC SUGGESTIONS.**

**a. Abstract line 5: delete 'similar'.**

*Authors:  Done*

**b. Abstract last sentence: move 'for precipitable water vapor (PWV) >10 mm,' to the start of the sentence.**

*Authors:  Done*

**c. All references citing 'World infrared standard group' should be replaced with 'World Infrared Standard Group' or after the first use WISG.**

*Authors: Done*

**d. Page 11 line 18: the ; before Nyeki et al should be replaced with 'and'**

*Authors:  Done*

**e. There are a number of other typos that one hopes and editor can correct.**

*Authors: An English-spoken proofreader has performed a detailed review of the manuscript, fixing the found typos.*

---

## Editor Comment (EC1) · S. Bekki (Editor) · 6 Apr 2018

1) Scientific Significance Does the manuscript represent a substantial contribution to modelling science within the scope of this journal (substantial new concepts, ideas, or methods)?

Good

2) Scientific Quality Are the scientific approach and applied methods valid? Are the results discussed in an appropriate and balanced way (consideration of related work, including appropriate references)? Do the models, technical advances and/or experiments described have the potential to perform calculations leading to significant scientific results?

Good

3) Scientific Reproducibility: To what extent is the modelling science reproducible? Is the description sufficiently complete and precise to allow reproduction of the science by fellow scientists (traceability of results)?

Excellent

4) Presentation Quality Are the scientific results and conclusions presented in a clear, concise, and well structured way (number and quality of figures/tables, appropriate use of English language)?

Excellent

For final publication, the manuscript should be

accepted subject to minor revisions

Please also note the supplement to this comment:
https://www.geosci-model-dev-discuss.net/gmd-2017-303/gmd-2017-303-EC1-supplement.pdf

**Supplement:**

This paper presents a comparison between observed and modelled LDR from 2010 and 2016 at the Izana station for both day-time and night-time. I really enjoyed reading this paper, which is quite clear and presents very interesting and important results for the community. I do recommend this paper for publication, but after some minor revisions. I have also some few questions that need to be addressed and could improve the paper. I have only one major comment about the paper regarding the way the authors are estimating the uncertainty of the models (section 4.2).

Specific comments/questions:

Page 1 lines 16-17: Are there any more recent reviews regarding the anthropogenic greenhouse gas?

Page 1 line 18: What are the uncertainties required on LDR measurements to assess completely their impact on climate changes?

Page 2 line 4: Could you specify how much represents 7.5 W. m-2 over the [4-42 micrometer] band?

Page 2 line 13: "Stefan-Boltzmann" is misspelled wrongly several times within the paper.

Page 4 Figure 1: Can you explain what the physical meaning of the upper and lower limit?

Page 4 lines 5-6: You are mentioning the calibration date of the instruments. How do these instruments degrade over time (% per year)? Has been any degradation characterization?

Page 5: There is no need for a 3.0.1 title here.

Page 5: from your regression study, it is possible to estimate the uncertainty about ε_AD.

Page 7 line 6: why do you consider the spectral range from 4 to 100 micrometer, when your spectral band of interest is [4-42] micrometer?

Page 7: I found the section 4.1 about the input parameters a bit confusing to follow, and I would suggest organizing things a bit more neat. The text would be easier to follow if the input parameters are presented as a clear list, with an equivalent structure.

Page 8 about the section 4.2: the method used by the author to estimate the uncertainty is rather simple, and is not statistically relevant and I would strongly advise the authors to change their method. A Monte-Carlo approach considering a normal distribution for each uncertainty parameters would definitely give a more appropriate estimate of the RT model global uncertainty. It requires however more computing time than the one run approach from the authors. In that case, the table 3 is no longer needed and can be replaced by a much smaller table resuming the global uncertainty using a Monte-Carlo approach.

Page 8 Table 3: the combined uncertainty for Modtran model is wrong (I guess a copy-paste from the combined uncertainty for the LibRadtran model).

Page 10 lines 6-7: Could you explain the main differences between the two models? It is not clear at all what makes these two RT models different, and the excellent comparison with the BSRN data does not shed much light about the RT performances differences.

Page 11 Table 4: Please change R -> r for the Pearson correlation coefficient. Why showing the correlation coefficient r and not $R^2$, which in your case of a linear regression is the square of r. $R^2$ represents the proportion of variance that can be explained by the linear regression. For day time with BSRN/LibRadtran, $R^2$ = 0.962 so 3.8 % of the variance is not explained by the linear regression. Also, why do the authors mentioned here the STD, since there is no reference within the text? What more information can bring the STD here when this clearly the RMSE that is valuable for the authors?

Page 11 line 10: I cannot find any mention of the LOWTRAN model earlier in the text. Please introduce it by explaining the differences with MODTRAN and LIBRADTRAN. Same for the SBDART model.

Page 12 line 21-22: I do understand that this outside the scope of this paper, but the authors should discuss which are the further ways or analysis needed to understand those discrepancies. This would indeed considerably make the paper more valuable other than just a data set description (although valuable for the community).

Page 12: The authors presented box plot of bias versus PWV. And what about the same results versus AOD (2nd uncertainty source)? Would the authors find also a clear pattern as for the PWV?

Page 13 Figure 5: What does represent the vertical dash line for January 2013? The authors mentioned in the text a "change point" for October 2012? I would also suggest here to add the histograms of the bias to have a better view of the distribution of the residuals. Assuming a normal distribution of the residuals, then the STD of table 4 would then be meaningful. But there is no reason that the distribution should be normal. Especially after what has been discussed with the impact of the PWV and the temporal stability.

---

## Author Comment (AC4) · 4 May 2018

**Dr. Slimane Bekki**

This paper presents a comparison between observed and modelled LDR from 2010 and 2016 at the Izana station for both day-time and night-time. I really enjoyed reading this paper, which is quite clear and presents very interesting and important results for the community. I do recommend this paper for publication, but after some minor revisions. I have also some few questions that need to be addressed and could improve the paper. I have only one major comment about the paper regarding the way the authors are estimating the uncertainty of the models (section 4.2).

Authors: We appreciate the positive feedback and constructive comments of the Editor. The major comment regarding the estimation of the models uncertainty is addressed hereinafter.

**Specific comments/questions:**

Page 1 lines 16-17: Are there any more recent reviews regarding the anthropogenic greenhouse gas?

Authors: Following the Editor's recommendation, the authors have added the following references: Iacono et al., (2008), Philipona et al., (2012), Wang and Dickinson (2013), Wild et al., (2013), Wild et al. (2015).

Page 1 line 18: What are the uncertainties required on LDR measurements to assess completely their impact on climate changes?

Authors: We have added the following paragraph to the final section of summary and conclusions:

"Considering that the BSRN measurement accuracy target for LDR is ±2 Wm-2, the average observed LDR change from 24 BSRN sites since early 1990s has been +2 Wm-2 dec-1 (Wild, 2017) as a result of the increase of the greenhouse effect, and the CMIP5 projections estimate LDR increases between 1.7 Wm-2 dec-1 (RCP4.5) and 2.2 Wm-2 dec-1 (RCP8.5) over the period 2010-2013 (Wild et al., 2015; 2017), it is crucial to ensure good consistency between LDR observations and estimates with models, such as the one found in this study. We can say that with the present LDR measurement accuracy, a period of

time less than two decades would be necessary for assessing completely its impact on climate change".

**Page 2 line 4: Could you specify how much represents 7.5 Wm-2 over the [4-42 micrometer] band?**

Authors: Note that the sentence "...The spectral range covers from 4 to 42  $\mu$ m with an expected sensitivity of 5 to 15  $\mu$ V/Wm-2, an uncertainty < 3% for daily totals, and an estimated inaccuracy < 7.5 Wm-2 (Kipp and Zonen, 2014)..." has been removed following the Referee#1's suggestion (see Response Referee #1).

Page 2 line 13: "Stefan-Boltzmann" is misspelled wrongly several times within the paper.

**Authors: Done.**

Page 4 Figure 1: Can you explain what the physical meaning of the upper and lower limit?

Authors: The upper and lower limits (Physically possible (PP): 40-700 W/m2 and Extremely rare (ER): 60-500 W/m2) correspond to black body temperature of -100°C and 60°C (Gilgen et al., 1995; Long and Dutton, 2002).

Page 4 lines 5-6: You are mentioning the calibration date of the instruments. How do these instruments degrade over time (% per year)? Has been any degradation characterization?

Authors: We appreciate this comment, whose answer is very timely to include in the manuscript.

Yes. We have estimated the degradation of the pyrgeometer using to consecutive calibrations performed in June 2014 and March 2017. The change in the calibration coefficients is 0.21% in 2.75 years, so the degradation is 0.08%/yr.

We have included the following text in Section 3.1, and modified Table 1 accordingly.

"In this study, we analyzed measurements performed with two CG(R)4 series (see Table 1) between 2010 and 2016 at IZO. The CG(R)4 #080022 was calibrated by the manufacturer in February 2008 at Holland (Kipp & Zonen) and the CG(R)4 #050783 was calibrated in June 2014 and March 2017 at the Physikalisch-Meteorologisches Observatorium Davos/World Radiation Center (PMOD/WRC). Given the two calibration coefficients of the second instrument (see Table 1), we estimate that its degradation is very small, lower than 0.08%/yr."

**Page 5: There is no need for a 3.0.1 title here.**

Authors: Done.

Page 5: from your regression study, it is possible to estimate the uncertainty about  $\varepsilon$ \_AD.

Authors: Following the Editor's recommendation, the authors have added the uncertainty of  $\epsilon_{AC}$ :  $\epsilon_{AC}$ =(0.218 ± 0.05)+(0.385 ± 0.07)x1/8

Page 7 line 6: why do you consider the spectral range from 4 to 100 micrometer, when your spectral band of interest is [4-42] micrometer?

Authors: The spectral response function of a pyrgeometer normally covers a wavelength range from 3.5 mm to about 50 mm. However, pyrgeometers are calibrated in the total range of terrestrial longwave emission (4 to 100  $\mu$ m).

Page 7: I found the section 4.1 about the input parameters a bit confusing to follow, and I would suggest organizing things a bit more neat. The text would be easier to follow if the input parameters are presented as a clear list, with an equivalent structure.

Authors: Following the Editor's recommendation (Pag 7 and Pag 10, 6-7), the authors have modified and clarified the Section 4 as follows:

**SECTION 4: RADIATIVE TRANSFER MODELS AND INPUT PARAMETERS**

"The simulations of surface LDR were determined with two RTMs: libRadtran and MODTRAN. The LibRadtran model (freely available from http://www.libradtran.org; Mayer and Kylling (2005)) used in this work is the version 2.0.1 (Emde et al., 2016). The simulations were performed with highly resolved absorption coefficients that were calculated using the absorption band parameterization called REPTRAN. It is based on the HITRAN 2004 spectroscopic database, in which wavelength-integrals have been parameterized as weighted means over representative wavelengths (Gasteiger et al., 2014). The simulations performed using REPTRAN in the thermal range showed relative differences of about 1% with respect 5 to simulations performed with high spectral resolution models and they are 6-7 times better than the simulations done with the LOWTRAN band parameterization (Gasteiger et al., 2014).

The MODTRAN version used in this work is the MODTRAN v6 (Berk and Hawes, 2017), an atmospheric transmittance and radiance model developed by the U. S. Air Force Research Laboratory in collaboration with Spectral Sciences, Inc. We have selected a band model with a resolution of 1 cm-1 for spectral calculations. The MODTRAN band model molecular spectroscopy is based on the High-resolution TRANsmission molecular absorption (HITRAN) database (Rothman et al., 2013).

For both models, the LDR simulations were calculated by using as radiative transfer equation (RTE) solver the DISORT (DIScrete ORdinate Radiative Transfer solvers), developed by Chandrasekhar (1960) and Stamnes et al. (1988, 2000), and based on the 5 multi-stream discrete ordinates algorithm. The number of

streams used to run Disort was 16. For each simulation, the integrated downward irradiance has been calculated in the spectral range 4-100 μm.

The two models were run using the same inputs, atmosphere and geometry in order to minimize.

The rest of the inputs measured at IZO are:

Radiosondes: Temperature and relative humidity (RH) profiles

In this work, we have used the AEMET's meteorological radiosondes dataset. Radiosondes are launched twice a day, at 11 and 23 UTC at the Güimar station (WMO GRUAN station #60018, 105 m a.s.l.). This station is located at the coastline, approximately 15 km to the southeast of IZO. Vertical profiles of pressure, temperature and relative humidity were obtained with Vaisala RS92 radiosondes (Rodríguez-Franco and Cuevas, 2013; Carrillo et al., 2016). We have used the radiosonde profiles from the altitude of IZO (2373 m a.s.l.).

• PWV

Since January 2009, the PWV has been obtained every 1h at IZO from a (Global Navigation Satellite System (GNSS) receiver considering satellite precise orbits (Romero Campos et al., 2009). In this work, we have used the PWV median measured between 11-13 and 23-01 UTC in order to take into account the radiosonde flight time, and hence making possible a comparison with GNSS observations.

• N2O and CO2 profiles

The volume mixing ratio (VMR) profiles of atmospheric CO2 and N2O trace gases were used. These were obtained from the monthly average profiles performed with the ground-based Fourier Transform InfraRed spectrometer (FTIR) at IZO between 1999 and 2015 (Schneider et al., 2005; García et al., 2014; Barthlott et al., 2015). The FTIR program at IZO is part of the Network for the Detection of Atmospheric Composition Change (NDACC). In this study FTIR climatological profiles have been used. The profiles were scaled on a daily basis with groundlevel in-situ CO2 and N2O mixing ratios, continuously measured at IZO since June 1984 and June 2007, respectively, within the WMO GAW programme (Cuevas et al., 2015, 2017).

• In-situ N2O and CO2

Since 2007 the  $CO_2$  in-situ measurements have been performed with a NDIR analyzer (LICOR-7000) (Gómez-Peláez and Ramos, 2009; Gómez-Peláez et al., 2010) and the N2O in-situ measurements with a VARIAN (GC-ECD 3800) (Scheel, 2009). We have used in this work only the night-time (20-08 UTC) averaged  $CO_2$ and N2O data because during this period IZO is under background free troposphere conditions, and the observatory is not affected by local and regional sources of such gases.

• AOD

Atmospheric aerosols have been included in the simulation process by means of the column-integrated aerosol optical depth (AOD), extracted from AERONET (Level 2.0 of version 2, cloud screened and quality ensured). AOD is obtained from solar observations performed with CIMEL sunphotometers at different wavelengths (Holben et al., 1998; Dubovik and King, 2000; Dubovik et al., 2006). The Shetle's aerosol model (Shettle, 1990) has been used in this study. The default properties are: rural type aerosol in the boundary layer, background aerosol above 2 km, spring-summer conditions and a visibility of 50 km. In this work, AOD at 500 nm has been used as model input. For day-time we have used the nearest AOD value to 11 UTC, and for night-time the last available AOD value of the day.

• Total ozone column (TOC)

TOC measurements with Brewer spectrometer began in 1991 at IZO. Since 2003 IZO has been appointed the Regional Brewer Calibration Center for Europe (RBCC-E; http://www.rbcc-e.org) and the total ozone program has been part of NDACC network. We have considered daily total ozone mean value as model input."

Page 8 about the section 4.2: the method used by the author to estimate the uncertainty is rather simple, and is not statistically relevant and I would strongly advise the authors to change their method. A Monte-Carlo approach considering a normal distribution for each uncertainty parameters would definitely give a more appropriate estimate of the RT model global uncertainty. It requires however more computing time than the one run approach from the authors. In that case, the table 3 is no longer needed and can be replaced by a much smaller table resuming the global uncertainty using a Monte-Carlo approach.

**Authors:** The authors have followed the "Guide to the Expression of Uncertainty in Measurement" (GUM) to estimate the RTM uncertainty in this work. This guide establishes general rules for evaluating and expressing uncertainty in measurement, and states the uncertainty can be evaluated by means of the statistical analysis of our observations (Type A evaluation). The combined standard uncertainty is therefore obtained from the positive square root of the sum of the different contributions. A least squares fitting has been performed to estimate these individual contributions. This approach was first applied by Rodgers (2000), and later is described in many references in the literature, (Schneider and Hase 2008; García et al. 2014, Schneider et al. 2006). Rodgers (2000) demonstrated that it is possible to separate systematic and random errors from a least squares fit of errors (Sim +  $\delta$ ) versus un-perturbed values (Sim). The slope of the regression fit can be identified as a systematic sensitivity error while the offset can be assumed as a systematic bias error. The random error is evaluated by means of the scatter around the regression line. Looking at these references, the authors consider that this method is statistically relevant despite being simpler than a Monte Carlo approach.

As Belluardo et al. (2016) claimed, fewer studies exist on how much the input uncertainty propagates into the radiative transfer code simulations. Monte Carlo technique seemed to them a more efficient way to study their uncertainty, but they also admitted there are other ways to

estimate these uncertainties in which possible correlations between these inputs parameters are properly taken into account.

We have seriously considered addressing an uncertainty analysis with the Monte Carlo method, as proposed by the Editor. However, an estimate of the number of simulations that are necessary to perform the RTM uncertainty with this approach shows us that we would need more than 4.000 model runs (2 model inputs, 500 trials, day-time, night-time, 2 RTMS, plus parameters combination) varying only the most important input parameters (PWV and AOD) since the contribution to the total uncertainty of the rest of the parameters is negligible. In addition to all this, and as has been answered to Referee#1, we could not address the inaccuracies in the observed temperature/humidity profiles due to different heating of the radiosonde sensors by solar radiation, which might be a clear uncertainty source. On the other hand, the agreement between RTMs is quite good (MB <1 Wm-2, 0.4%), very similar to the uncertainty estimated for both RTMs (<0.95 Wm-2, 0.5%) which suggests that the estimation of the uncertainty estimated with the GUM approach is reasonable.

These results don't justify a more complex calculation of the uncertainty with a computational cost so high. An analysis with the Monte Carlo Method would constitute a study (and a publication) itself, and we cannot address it in this study.

Page 8 Table 3: the combined uncertainty for Modtran model is wrong (I guess a copy-paste from the combined uncertainty for the LibRadtran model).

**Authors: Corrected.**

Page 10 lines 6-7: Could you explain the main differences between the two models? It is not clear at all what makes these two RT models different, and the excellent comparison with the BSRN data does not shed much light about the RT performances differences.

Authors: We have included the following text into the manuscript:

"The main differences between the two models is in the molecular absorption band: while MODTRAN uses High-resolution TRANsmission molecular absorption (HITRAN) database (Rothman et al., 2013), LibRadtran uses the absorption band parameterization called REPTRAN (Gasteiger et al., 2014)."

Page 11 Table 4: Please change R -> r for the Pearson correlation coefficient. Why showing the correlation coefficient r and not R2, which in your case of a linear regression is the square of r. R2 represents the proportion of variance that can be explained by the linear regression. For day time with BSRN/LibRadtran, R2 = 0.962 so 3.8 % of the variance is not explained by the linear regression. Also, why do the authors mentioned here the STD, since there is no reference within the text? What more information can bring the STD here when this clearly the RMSE that is valuable for the authors?

Authors: Following the Editor's recommendation, the authors have removed STD from Table 4 and have used R2 in the manuscript.

Page 11 line 10: I cannot find any mention of the LOWTRAN model earlier in the text. Please introduce it by explaining the differences with MODTRAN and LIBRADTRAN. Same for the SBDART model.

Authors: References to both LOWTRAN and SBDART are done in the introduction of the manuscript (page 2). However, the authors do not consider it appropriate to describe LOWTRAN and SBDART since these RTMs are not used in this study.

Page 12 line 21-22: I do understand that this outside the scope of this paper, but the authors should discuss which are the further ways or analysis needed to understand those discrepancies. This would indeed considerably make the paper more valuable other than just a data set description (although valuable for the community).

Authors: The authors have added more information following the recommendation of referee 1 (see Response Referee #1) (see final manuscript uploaded on March 26):

"...The small differences observed in the evolution of the LDR bias with the PWV (close to the instrumental error) found between day-time and night-time are not currently understood. It is likely that this different behaviour between day and night may be associated with instrumental measurements (Ohmura et al., 1998; McArthur, 2005), but we do not preclude they could be also related to inaccuracies in the model input parameters during day-time, e.g., inaccuracies in the observed temperature/humidity profiles due to different heating of the radiosonde sensors by solar radiation. Dirksen et al. (2014) studied the effects on the RS92's temperature and humidity measurements and they estimated this uncertainty to be 0.15 K for night-time temperature measurements and approximately 0.6 K at 25 km during daytime..."

Page 12: The authors presented box plot of bias versus PWV. And what about the same results versus AOD (2nd uncertainty source)? Would the authors find also a clear pattern as for the PWV?

Authors: The authors appreciate this suggestion. Specific analysis of bias dependence on AOD and PWV has been included in the paper as follows:

According to the results obtained in Section 4.2, the uncertainties on PWV and AOD dominate the total uncertainty, thus, the LDR bias have been analyzed.

The box plot of LDR bias for different PWV is presented in Figures 4a and 4b. Both models tend to underestimate LDR (up to 5 Wm-2) in the case of day-time measurements with PWV<9 mm (Figure 4a). A LDR bias around zero is observed for higher PWV, although it is necessary to emphasize that the number of data in this PWV range (between 4% and 5%) is much lower. At night-time, the dependence of LDR bias with PWV shows a negligible bias under dry conditions (PWV<6 mm), and a slight overestimation of both models (up to +5 Wm-2) for higher PWV values (Figure 4b). These results are consistent with those obtained by Gröbner et al. (2014) and Nyeki et al. (2017) which argue that the World Infrared Standard Group (WISG) of pyrgeometers has a negative bias of about 5 Wm-2 under cloud-free conditions and PWV>10 mm.

---

## Author Response (AR2)

**Anonymous Referee #1:**

**GENERAL COMMENTS**

The manuscript presents a comparison of calculated and observed longwave downward radiation (LDR) for cloud-free conditions at the BSRN Izaña station using the radiative transfer models libRadtran and MODTRAN. Differences(bias ±RMSE)between calculated and observed LDR for 1014 night-time cases in the 2010-2016 period were less than5 Wm-2and hence within the measurement uncertainty with the model calculations being higher compared to the observations. Discrepancies between models and observationsfor1048 cloud-free day cases were slightly higher with the models yielding lower irradiances. These differences in the statistics between day and night are currently not yet understood. Finally, the models confirm the water vapour dependency of observations traceable to the World Infrared Standard Group (WISG) which has been postulated in earlier studies using windowless radiometers (e.g., Gröbner et al., 2014).

The LDR is the second largest component in the radiation budget, directly related to the greenhouse effect and hence of great importance. The accurate calculation of the cloud-free LDR is relevant in many applications. Therefore, the manuscript is significant for the community and hence appropriate for this journal. The manuscripts in general well-structured and clearly written. Graphics and tables are clear and the captions self-explanatory. Therefore, I would recommend publishing the article in GMD after minor-mainly technical -revisions.

Authors: We appreciate the positive and constructive comments of the Referee. Here we discuss and respond to his/her specific comments and technical corrections.

**2.) SPECIFIC COMMENTS**

**i) Cloud-free detection:**

p.5: I wonder if no observational method for night time is available at this site which detects high level clouds more reliably than the Clear-Sky Index (CSI) or APCADA does. Can you comment on that? Moreover, did you really use APCADA (i.e., did you determine the diurnal and annual variability of k and did you calculate fractional cloud cover) or did you just use the original CSI from Marty and Philipona (2000) which only distinguishes between cloud-free and cloudy skies? Please specify. It seems to me that you used the CSI from Marty and Philipona (2000) if this is true delete APCADA and the corresponding reference in the text.

Authors: The Izaña Observatory has a SONA camera (Automatic Cloud Observation System; González et al., 2013) which takes an image every 5 minutes during all day. These images have helped us to check the cloud-free results obtained with the Clear-Sky Index and Long and Ackerman method. We have reviewed the images with a visual examination during the daytime (11 UTC) and night-time (23 UTC), but this information was not incorporated into the original manuscript and will be added in the final manuscript.

The referee is right and the method used to detect cloud-free and cloud-skies was the one developed by Marty and Philipona (2000) and not the APCADA (Dürr and Philipona, 2004).

For this reason, the section "Cloud-free detection" has been modified as follows:

"The cloud-free days were detected by using the algorithm developed by Marty and Philipona (2000). In this algorithm, a Clear-Sky Index (CSI) is calculated to separate cloud-free from cloud-sky situations using accurate measurements of LDR in conjunction with air temperature and relative humidity values measured at the station. The CSI index is defined as:

$$CSI = \frac{\varepsilon_A}{\varepsilon_{AC}} \tag{2}$$

where

$$\varepsilon_A = \frac{LDR}{\sigma_b T^4} \tag{3}$$

$$\varepsilon_{AC} = \varepsilon_{AD} + k(e/T)^{1/8}$$
(4)

where  $\sigma_b$  is the Stefan-Boltzmann constant, T is the air temperature (K),  $\varepsilon_{AD}$  is an altitude-dependent emittance of a completely dry atmosphere, e is water vapor pressure (Pa) and k is a constant coefficient dependent on the location. If CSI Index  $\leq$  1 indicates cloud-free (no clouds) and if CSI Index >1 indicates cloud-sky (overcast) (Marty and Philipona, 2000).

In order to calculate  $\varepsilon_{AC}$  this method requires the evaluation of  $\varepsilon_{AD}$  and k, as shown in equation (4). A sample of known cloud-free days is used to plot  $\varepsilon_{AC}$  against  $e'_T$ (Figure 2). The cloud-free condition of this sample is assured by applying the Long and Ackerman's method (Long and Ackerman, 2000; adapted for IZO by García et al. (2014)). This method is based on surface measurements of global and diffuse solar radiation with a 1-min sampling period and consists of in four individual tests applied to normalized global radiation magnitude, maximum diffuse radiation, change in global radiation with time, and normalized diffuse radiation ratio variability. We have considered the period 2010-2016 at 11 UTC to determine the fitting coefficients of equation (4) obtaining the following relationship (Figure 2):  $\varepsilon_{AC} = 0.218 + 0.385 (e/T)^{1/8}$

Despite the  $\varepsilon_{AD}$  depends on the altitude of the station we have obtained for IZO a value of 0.218, similar to the values obtained by Marty and Philipona (2000) for stations located between 2230 and 2540 m (0.22 and 0.211, respectively).

Once we have adjusted the coefficients, the cloud-free cases were selected with a combination of Long and Ackerman and CSI methods. At day-time, we have used the Long and Ackerman one, taking into account for each day the period 11-13 UTC. At night-time the CSI was applied in the period 23-01 UTC. These results have been checked with a visual examination of the SONA camera (Automatic Cloud Observation System; González et al., 2013) installed at IZO. We found that both methods detect 97% of the visually selected. In the period 2010-2016 a total of 1161 and 1083 cases were detected, for day-time and night-time respectively. "

ii) Solar effect on the LDR and differences in the bias between the day and night comparisons of observed and calculated LDR:

I assume that the LDR observations used in this study were shaded (according to the guidelines of BSRN), i.e. both pyrgeometers were installed on a solar tracker? It is a bit confusing because the authors state (on p.4, line4, based on McArthur (2005)) that the CG4 filters all solar radiation and hence no shading is necessary (I see this statement anyways a bit more critical: the longwave irradiance in the direct beam of the sun is measured by any pyrgeometer and its magnitude depends on the cut-on of the filter and the solar insolation and hence on atmospheric conditions (e.g., water vapor content, cloudiness).In fact, a CG(R)4 has a higher cut-on (approximately at 4.5 $\mu$ m) compared to a Eppley PIR (approximately at 4 $\mu$ m)and hence the CG(R)4 measures less longwave irradiance from the sun which has been already reported in previous studies (e.g., Meloni et al., 2012). Nevertheless, a few Wm-2 originating from the long-wave irradiance in the direct beam of the sun will be observed by a CG(R) 4 and thus it should be also operated in shaded mode).In any case, state clearly if your pyrgeometers were shaded (e.g., on p.3, line 30: '...with a shaded and ventilated broadband Kipp & Zonen...', or on page 4, line 4 after the reference of McArthur (2005)).

If the pyrgeometers were not shaded (unlikely), the long-wave irradiance in the direct beam of the sun could be a possible explanation for the small differences in the bias between the results of the day and night comparisons of measured and calculated LDR (p.11, lines20-21/p.12, lines 1-2and Fig. 4 or Table 4) which are in fact consistent with the results in Dürr et al, 2005. If the observations are shaded, it is reasonable that the differences between day and night are caused by additional measurement inaccuracies during daytime as stated by the authors. However, an underestimation of the models due to inaccuracies in the model input parameters during day time (e.g., inaccuracies in the observed temperature/humidity profiles due to different heating of the radiosonde sensors by solar radiation) could be also possible (instead of instrumental inaccuracies). Could you comment on that? I would add this option at the end of the paragraph (p.12, line 2).

(5)

Authors: The authors appreciate your interesting comments. The LDR observations have been performed with CGR4 pyrgeometers installed on a solar tracker and shaded. This information has been added to the final manuscript in Pag. 3 Line 30 as follows:

"The LDR measurements used in this study have been performed at the Izaña BSRN (#61, IZA; http://www.bsrn.aemet.es) (García et al., 2012) with a broadband Kipp & Zonen CG4 pyrgeometer (onwards, CGR4) mounted on a sun tracker equipped with a shadowing and ventilated. This instrument uses a specially designed silicon window which, provides a 180° field of view (although not hemispherical) with good cosine response. A diamond-like surface protects the outer surface of the window, while the inner surface filters all solar radiation. The design of the instrument is such that solar radiation absorbed by the windows is conducted away to reduce the solar heating effect. This fact reduces the need for dome heating correction terms and shading from the sun (McArthur, 2005)."

**Pag 12 Line 2:**

"The small differences observed in the evolution of the bias with the PWV (close to the instrumental error) found between day-time and night-time may be associated with instrumental measurements (Ohmura et al., 1998; McArthur, 2005) and they could also be related to inaccuracies in the model input parameters during day-time, e.g., inaccuracies in the observed temperature/humidity profiles due to different heating of the radiosonde sensors by solar radiation. Dirksen et al. (2014) studied the effects on the RS92's temperature and humidity measurements and they estimated this uncertainty to be 0.15 K for night-time temperature measurements and approximately 0.6 K at 25 km during daytime. "

**Pag 14 Line 4:**

"The differences between day and night-time are currently not yet understood, therefore further specific analysis is needed to understand these differences which is outside the scope of this work."

**SUMMARY:**

p.3, line 30 or p.4, line 4: Specify if the LDR observations were shaded or not.

Authors: Following the Referee's recommendation, the authors have added that the pyrgeometer is shaded (See previous answer (SPECIFIC COMMENTS ii).

p.4, line 2: Replace 'all solar radiation' by 'most of the solar radiation' *Authors: Done.*

p.11, line 13-21: I would re-arrange this paragraph and start with the night-time results first, i.e. with line 16 (in the night the uncertainties are in general smaller because of the absence of solar radiation). Then describe the results for day time.

Authors: The authors think it is not convenient to discuss night-time results before day-time, for consistency with the rest of the manuscript.

p.11,lines20-21/p.12, lines 1-2: If the LDR observations were not shaded the previously mentioned impact of the longwave irradiance in the solar spectrum on the LDR observations should be stated and the publication of Meloni et al. (2012) cited. If the observations are shaded, I agree with the content (but I would use '...with additional instrumental inaccuracies during daytime' on p.11 line 21/p.12, line 1). In addition, I would add a sentence about possible inaccuracies in the model input parameters during day time which may result in an underestimation of the models.

Authors: See previous answer (SPECIFIC COMMENTS ii).

**3.) TECHNICAL CORRECTIONS**

p.1, title: add 'cloud-free' between 'modelled' and 'longwave'. Authors: Done.

p.1, line 4 and throughout the manuscript: 'libRadtran' instead of 'LibRadtran'. *Authors: Done.*

p.1, line 4: Revise sentence: 'Results show an excellent....and simulations using the radiative transfer models (RTM) libRadtran and MODTRAN V6.'( delete 'similar for both models'). *Authors: Done.*

p.1, line 7: '...useful tools for the quality control of LDR observations...' Authors: Done.

p.1, line 16: cloud cover is only one aspect; I would add 'cloud type'. Furthermore, water vapor is missing.

Authors: Following the Referee's recommendation, the authors have been modified this added this sentence:

"The longwave downward radiation (LDR) at the Earth's surface is a key component in land-atmosphere interaction processes, and is crucial in the surface energy budget and global climate change, because the changes in the LDR values may be related to changes in cloud-cover, cloud type, water vapour, temperature, and the increase of anthropogenic greenhouse gas concentrations in the atmosphere (Wild et al., 1997; Marty et al., 2003)."

p.2, line 2: The CG4 is nowadays termed CGR4 use 'CG(R)4 series' Authors: Done.

p.2, line 2: put the reference of McArthur (2005) at the end of the sentence. *Authors: Done*.

p.2, lines 3-4. I would delete this sentence. The specifications for the CG(R)4 from Kipp & Zonen may not be representative for the other types of pyrgeometers listed previously.

**Authors: Done.**

p.2, line 6: delete here the reference of Ohmura et al. (1998). Authors: Done.

p.2, line 9: Reference should be Ångström, also in the reference list. Authors: Done.

p.2, line 11, use 'e.g.,' instead of 'i.e.' Authors: Done.

p.2, line 17, 'Stefan-Boltzmann constant' Authors: Done.

p.2, line 27: I would put 'as model inputs' at the end of the sentence. *Authors: Done.*

p.3, line 28: I would term Section 3 as 'Observational Data and Methods', then Section 3.1 'Instrument and Measurements' and Section 3.2'Cloud-free detection' Authors: Done.

p.3, line 4: '...with values of +1.5and –3.2Wm-2for night-time....' Authors: Done.

p.3, line 12: rather use'...uncertainty assessment...'than '...quality assessment...' Authors: Done.

p.3, line 13: '...temporal stability of the LDR observations...' Authors: Done.

p.3, line 17: use 'location' instead of 'situation'. Authors: Done.

p.3, line21: I would use '...it has been actively contributing...' *Authors: Done.*

p.3, line 21 and throughout the manuscript: an abbreviation should be define at its first occurrence in the manuscript, e.g., '...such as the Network for the Detection of Atmospheric Composition Change(NDACC; http://www.ndsc.ncep.noaa.gov/)since 1999,the Aerosol Robotic Network(AERONET, http://aeronet.gsfc.nasa.gov/) since 2004, the Total Carbon Column Observing Network (TCCON, http://www.tccon.caltech.edu/)since2007,...'. Later, just use the abbreviation.

**Authors: Done.**

p.3, line 26: Revise reference (also in the reference list). Should be read 'WMO' or 'CIMO', I guess.

Authors: Done.

p.4, line 6: '...at the Physikalisch-Meteorologisches Observatorium Davos/World Radiation Center (PMOD/WRC).' *Authors: Done.*

p.4, line 9: The reference is from 2002, I guess. Revise also in the reference list.

**Authors: Done.**

p.5, line 1: Here, I would use only the reference of Dürr and Philipona (2004) but only if you have really used APCACA (see my previous comments). Insert the reference of Marty and Philipona inline 4 (after 'at the station'). If you have used the CSI from Marty and Philipona (2000) replace APCADA and the corresponding reference with 'Clear-Sky Index (CSI) (Marty and Philipona(2000)'in line 1, p.5.

Authors: Done. See previous answer (SPECIFIC COMMENTS i).

p.5, line 10: 'Stefan-Boltzmann' Authors: Done.

p.5, line 10:  $\epsilon$ AD is an altitude-dependent emittance of a completely dry atmosphere ( $\epsilon$ AC is the apparent emittance of a cloud-free sky)

Authors: Done. See previous answer (SPECIFIC COMMENTS i).

p.5, lines11/12: Revise this sentence, e.g.: 'A CSI Index ≤1 and >1indicates cloud-free and cloud-sky, respectively.'

Authors: Done. See previous answer (SPECIFIC COMMENTS i).

p.5, line 16: '...consists of...' Authors: Done.

p.5, line 28: delete 'models'.

Authors: Done.

p.6, line 7: Hasn't the band model used in MODTRAN 6 a resolution of 0.1cm-1?

Authors: Yes, the MODTRAN 6 has resolution of  $0.1 \text{ cm}^{-1}$ , but we use a resolution of  $1 \text{ cm}^{-1}$  in order to reduce de computational time, taking into account that the integrate LDR using 1 cm-1 resolution differs not significantly from the  $0.1 \text{ cm}^{-1}$  one.

p.7, line 10: The site of the radiosonde launch is located at sea level, more than 2000 m lower the IZO. I assume that you cut the profiles at the altitude of IZO to assimilate the profiles into the RTM?

Authors: Yes, the radiosonde profiles have been considered from the altitude of IZO (2400 m a.s.l.). The authors have been added this sentence:

"In this work, we have used the meteorological radiosondes dataset from IARC-AEMET. Radiosonde profiles have a temporal resolution of 12 h (at 11 and 23 UTC) and were launched at Güimar station (WMO GRUAN station #60018, 105 m a.s.l.). This station is located at the coastline, approximately 15 km to the southeast of IZO. Vertical profiles of pressure, temperature and relative humidity were measured using Vaisala RS92 radiosondes (Cuevas et al., 2015; Carrillo et al., 2016). We have considered the radiosonde profiles from the altitude of IZO (2373 m .a.s.l.)"

p. 7, line 18: NDACC has been already defined on p. 3.

**Authors: Done.**

p.7, line 31: delete 'one'.

**Authors: Done.**

p.8, line 15: delete 'the'. Authors: Done.

p.10, line 5: Did you average the observations over a certain time period (e.g., 30 minutes) in order to validate the RTM calculations? Or did you use the 1 min observations? Specify.

Authors: Yes, we have averaged in a 30 minutes period the LDR observations from 11:00 to 11:30 and 23:00 to 23:30 UTC to coincide with the launch of the radiosonde. This information has been added to the final manuscript as follows:

"In this section, we present the comparison between LDR measured with BSRN and simulated with LibRadtran and MODTRAN, considering the available and coincident cloud-free BSRN at day-time and night-time, and the inputs indicated in section 4.1 at IZO between 2010 and 2016. A total of 1048 measurements at day-time, and 1014 measurements at night-time were used. The observations were averaged in a time period of 30 minutes, in order to agree with the mean duration of the radiosonde launch. In particular, we averaged over 11:00-11:30 UTC and 23:00-23:30 UTC periods, for day-time and night-time measurements respectively. ..."

p.10, line 7: you may better use '...and R of 0.999, and are more consistent during nighttime'.

**Authors: Done**

p.11, Table 4 (Caption): The number of day-time calculations given here (1075 cases) is not consistent with those given in the abstract, Section 5 (1048, p.10) and Section 6.

**Authors: Done**

p.12, line 11: Could you specify what was changed in the location of the instrumentation in 2012?

Authors: Due to improvement works at IZO the instrumentation was moved for a short period of time (approximately on month) from the tower to a platform on the ground. Once the works ended the instrumentation was moved to its original location. This information has been added to the final manuscript as follows:

"... When analyzing the BSRN LDR and the simulated LDR times series separately, we do not observe any change in the simulated LDR, but a change point in the BSRN LDR time series at both day-time and night-time. This change point (October 2012) coincided with a change in the location of the instrumentation within the IZO facilities, exactly the instrument was moved during a month from the tower to the ground until the works ended...."

**p.14, line 3: 'supports'.**

**Authors: Done**

p.14, line 4: I would add 'However, the differences between day and night are currently not yet understood.'

Authors: The authors have been added this sentence to the final manuscript (See previous answer (SPECIFIC COMMENTS ii).

p.17, line 37: Specify journal/meeting event of publication/presentation of Redondas and Cede.

Authors: Done

**GMD-2017-303**
The manuscript presents a concise comparison of 7 years of downward longwave radiation measurements obtained at the Izaña Atmospheric Observatory to two high resolution radiative models using other measured parameters at the site. The results show agreement between the two models and pyrgeometer measurements to within their demonstrated uncertainties. This manuscript only needs minor adjustments for publication and will be of great benefit to both the modelling and measurement communities.

Authors: The authors acknowledge the referee's constructive comments, and in the followings, we discuss and respond to the general comments and specific suggestions.

**GENERAL COMENTS**

1. There is no indication of what DLR measureands are used in the comparison. Are they single sample, minute averages or longer averages. There is reference to 'instantaneous' measurements but such measurements do not exist as most data acquisition systems integrate over a small but finite period. For example, there is reference to 1-minute surface measurements in 3.0.1 but are they averages or single samples.

Authors: The authors have averaged in a 30 minutes period the LDR observations from 11:00 to 11:30 and 23:00 to 23:30 UTC to coincide with the launch of the radiosonde. This information has been added to the final manuscript as follows:

"In this section, we present the comparison between LDR measured with BSRN and simulated with libRadtran and MODTRAN, considering the available and coincident cloud-free BSRN at day-time and night-time, and the inputs indicated in section 4.1 at IZO between 2010 and 2016. A total of 1048 measurements at day-time, and 1014 measurements at night-time were used. The observations were averaged in a time period of 30 minutes, in order to agree with the mean duration of the radiosonde launch. In particular, we averaged over 11:00-11:30 UTC and 23:00-23:30 UTC periods, for day-time and night-time measurements respectively..."

2. On occasions 'accuracies' are given a quantitative value. In ISO accuracies are a qualitative (good, bad, indifferent) not quantitative. Just because a manufacturer incorrectly uses accuracy as a quantitative term is no reason to repeat bad practise.

Authors: The authors agree with this comment. We have replaced the following uses of accuracy:

Page 2, Line 4:

"...The spectral range covers from 4 to 42 \_m with an expected sensitivity of 5 to 15  $\mu$ V/Wm-2, an uncertainty < 3% for daily totals, and uncertainty < 7.5 Wm-2"

Page 2, Line 18:

*"...The above mentioned parameterizations show uncertainties ranging from 9% to 15 % in low altitude sites..."*

3. 'Temporal resolution' and 'temporal frequency' are used in 4.1 lines 7 to 15 - but what one thinks is meant is sampling rate.

Authors: Following the referee's recommendation, we have modified the two sentences as follows:

*Line 7:*

*In this work, we have used the meteorological radiosondes dataset from IARC-AEMET. Radiosonde profiles launched at 11 and 23 UTC in the Güimar station*

Line 15:

Since January 2009, the PWV has been obtained at IZO from a GNSS (GPS-GLONASS) receiver considering GPS precise orbits every 1 h (Romero Campos et al., 2009)

4. While the AOD at 500 nm is used there is no indication of the aerosol model (i.e. distribution) that scales in the IR.

Authors: Following the referee's recommendation, we have added this information in the final manuscript as follows:

"The atmospheric aerosols are included in the simulation process by means of the column-integrated AOD, extracted from AERONET (Level 2.0 of version 2, cloud screened and quality ensured). The AOD is obtained from solar observations performed with CIMEL sunphotometers at different wavelengths (Holben et al., 1998; Dubovik and King, 2000; Dubovik et al., 2006). The aerosol model used in this work is of Shetle (Shetle, 1989). The default properties are a rural type aerosol in the boundary layer, background aerosol above 2 km, spring-summer conditions and a visibility of 50 km. In this work, we have used AOD at 500 nm as model input. For day-time we have used the nearest AOD value to the 30 UTC, and for night-time the last AOD value of the day."

Shettle, E.: Models of aerosols, clouds and precipitation for atmospheric propagation studies, in: Atmospheric propagation in the uv, visible, ir and mm-region and related system aspects, no. 454 in AGARD Conference Proceedings, 1989.

5. Figure 3 shows a standard X vs Y plot of various comparison parameters. It would be more instructive as (Y-X) vs X plots with a (Y-X) = zero line.

Authors: We have made the Figure 3, following the referee's recommendation, however, the authors think that as this figure is similar to the Figure 5 of original manuscript, and therefore we think it more convenient to keep Figure 3 of the original manuscript.

Figure.- Difference between LDR (Wm-2) simulations with libRadtran (blue color) and LDR BSRN (Wm-2) at cloud-free (a) day-time and (d) night-time. Difference between MODTRAN LDR (Wm-2) (red color) and BSRN LDR (Wm-2) at (b) day-time and (e) night-time. Difference between MODTRAN LDR (Wm-2) and libRadtran LDR ((Wm-2) (black color) at (c) day-time and (f) night-time

6. Table 4. Unless one of the variables is the 'truth' then the RSME are really root mean square differences.

Authors: Thank you very much for the comment, the authors have been changed the root mean square error (RMSE) by root mean square of the bias (RMS) in the final manuscript.

7. 5.1.1 - while the step jump on relocation was detected there does not appear to be any comment on the different pygeometers. Was one replaced with another? If not, see point 1 above as it is not clear what measurements were used; a mean between the two?? If one was replaced with another then it would be worth saying that no jump in differences were detected when replacing an instrument.

Authors: The pygeometer was not replaced by any other. The instrumentation was moved for a short period of time (approximately one month) from the tower to a platform on the ground. Once the works ended the instrumentation was moved to its original location, without ceasing to measure at any time.

"... When analyzing the BSRN LDR and the simulated LDR times series separately, we do not observe any change in the simulated LDR, but a change point in the BSRN LDR time series at both day-time and night-time. This change point (October 2012) coincided with a change in the location of the instrumentation within the IZO facilities, exactly the instrument was moved during a month from the tower to the ground until the works ended...."

8. Section 6 line 1-5: 'suggest a scale change of the WISG' - this is an erroneous statement as the WISG is an interim scale until a better one can be found. It might be better to rephrase it to 'The support previous measurement studies that suggest an offset of the WISG to the SI.'

Authors: Following the referee's recommendation, we have modified this information in the final manuscript as follows:

"...The observed night-time difference between models and measurements of +5 Wm-2 for PWV> 10 mm supports previous measurements studies that suggest of the World Infrared Standard Group of Pyrgeometers (WISG)..."

**SPECIFIC SUGGESTIONS.**

a. Abstract line 5: delete 'similar'.

**Authors: Done**

b. Abstract last sentence: move 'for precipitable water vapor (PWV) >10 mm,' to the start of the sentence.

**Authors: Done**

c. All references citing 'World infrared standard group' should be replaced with 'World Infrared Standard Group' or after the first use WISG.

**Authors: Done**

d. Page 11 line 18: the ; before Nyeki et al should be replaced with 'and'

**Authors: Done**

e. There are a number of other typos that one hopes and editor can correct.

**Authors: Done**

**GMD-2017-303**
GMD does not necessary require for a Model evaluation paper to make statements about code availability. This is applied under the assumption that the manuscript is referencing a paper that describes the model being evaluated and that this paper states how to obtain access to the program code of the model. As this does not apply for this manuscript, the authors need to explain how to access the code. As stated in https://www.geoscientific-modeldevelopment.net/about/manuscript\_types.html for "Model description papers" the preferred option is that authors upload their code and the data as supplement.

Authors: The authors appreciate the comments.

In this work we present the comparison between measured and simulated longwave downward radiation. We have used two radiative transfer models: LibRadtran and MODTRAN. The LibRadtran model is free and available on the web: http://www.libradtran.org; Mayer and Kylling (2005) (This information is included in the manuscript Page 6 Line 1-7), while that the MODTRAN model is only available under the commercial agreement with Spectral Sciences, Inc. (http://modtran.spectral.com; Berk et al., 2000, 2008, 2013, 2015; Berk and Hawes, 2017).

For this reason, we consider that it is not possible to add the code for both models since these two models have not been developed by the authors.

**References**:

- 1. Berk, A., Acharya, P. K., Anderson, G., Chetwynd, J. H., and Hoke, M. L.: Reformulation of the MODTRAN band model for higher spectral resolution, in: Proceedings spue the international society for opitcal engineering, pp. 190–198, International Society for Optical Engineering; 1999, 2000.
- 2. Berk, A., P.K. Acharya, L.S. Bernstein, G.P. Anderson, P. Lewis, J.H. Chetwynd, and M.L. Hoke, "Band Model Method for Modeling Atmospheric Propagation at Arbitrarily Fine Spectral Resolution," U.S. Patent #7433806, issued October 7, 2008.
- 3. Berk, P. Conforti, R. Kennett, T. Perkins, F. Hawes, and J. van den Bosch, "MODTRAN6: a major upgrade of the MODTRAN radiative transfer code," Proc. SPIE 9088, Algorithms and Technologies for Multispectral, Hyperspectral, and Ultraspectral Imagery XX, 90880H (June 13, 2014); doi:10.1117/12.2050433.

- 4. Alexander Berk, Patrick Conforti, and Fred Hawes, "An accelerated line-by-line option for MODTRAN combining on-the-fly generation of line center absorption with 0.1 cm-1 bins and pre-computed line tails," Proc. SPIE 9471, Algorithms and Technologies for Multispectral, Hyperspectral, and Ultraspectral Imagery XXI, 947217 (May 21, 2015); doi:10.1117/12.2177444
- 5. Berk, A. and Hawes, F.: Validation of MODTRAN® 6 and its line-by-line algorithm, Journal of Quantitative Spectroscopy and Radiative Transfer, 203, 542–556, 2017.
- Mayer, B. and Kylling, A.: Technical note: The libRadtran software package for radiative transfer calculations – description and examples of use, Atmospheric Chemistry and Physics, 5, 1855–1877, https://doi.org/10.5194/acp-5-1855-2005, http://www.atmos-chemphys.net/5/1855/2005/, 2005.

**GMD-2017-303**
This paper presents a comparison between observed and modelled LDR from 2010 and 2016 at the Izana station for both day-time and night-time. I really enjoyed reading this paper, which is quite clear and presents very interesting and important results for the community. I do recommend this paper for publication, but after some minor revisions. I have also some few questions that need to be addressed and could improve the paper. I have only one major comment about the paper regarding the way the authors are estimating the uncertainty of the models (section 4.2).

Authors: We appreciate the positive feedback and constructive comments of the Editor. The major comment regarding the estimation of the models uncertainty is addressed hereinafter.

**Specific comments/questions:**

Page 1 lines 16-17: Are there any more recent reviews regarding the anthropogenic greenhouse gas?

Authors: Following the Editor's recommendation, the authors have added the following references: Iacono et al., (2008), Philipona et al., (2012), Wang and Dickinson (2013), Wild et al., (2013), Wild et al. (2015).

Page 1 line 18: What are the uncertainties required on LDR measurements to assess completely their impact on climate changes?

Authors: We have added the following paragraph to the final section of summary and conclusions:

"Considering that the BSRN measurement accuracy target for LDR is ±2 Wm-2, the average observed LDR change from 24 BSRN sites since early 1990s has been +2 Wm-2 dec-1 (Wild, 2017) as a result of the increase of the greenhouse effect, and the CMIP5 projections estimate LDR increases between 1.7 Wm-2 dec-1(RCP4.5) and 2.2 Wm-2 dec-1 (RCP8.5) over the period 2010-2013 (Wild et al., 2015; 2017), it is crucial to ensure good consistency between LDR observations and estimates with models, such as the one found in this study. We can say that with the present LDR measurement accuracy, a period of

time less than two decades would be necessary for assessing completely its impact on climate change".

**Page 2 line 4: Could you specify how much represents 7.5 Wm-2 over the [4-42 micrometer] band?**

Authors: Note that the sentence "...The spectral range covers from 4 to 42  $\mu$ m with an expected sensitivity of 5 to 15  $\mu$ V/Wm-2, an uncertainty < 3% for daily totals, and an estimated inaccuracy < 7.5 Wm-2 (Kipp and Zonen, 2014)..." has been removed following the Referee#1's suggestion (see Response Referee #1).

Page 2 line 13: "Stefan-Boltzmann" is misspelled wrongly several times within the paper.

**Authors: Done.**

Page 4 Figure 1: Can you explain what the physical meaning of the upper and lower limit?

Authors: The upper and lower limits (Physically possible (PP): 40-700 W/m2 and Extremely rare (ER): 60-500 W/m2) correspond to black body temperature of -100°C and 60°C (Gilgen et al., 1995; Long and Dutton, 2002).

Page 4 lines 5-6: You are mentioning the calibration date of the instruments. How do these instruments degrade over time (% per year)? Has been any degradation characterization?

Authors: We appreciate this comment, whose answer is very timely to include in the manuscript.

Yes. We have estimated the degradation of the pyrgeometer using to consecutive calibrations performed in June 2014 and March 2017. The change in the calibration coefficients is 0.21% in 2.75 years, so the degradation is 0.08%/yr.

We have included the following text in Section 3.1, and modified Table 1 accordingly.

"In this study, we analyzed measurements performed with two CG(R)4 series (see Table 1) between 2010 and 2016 at IZO. The CG(R)4 #080022 was calibrated by the manufacturer in February 2008 at Holland (Kipp & Zonen) and the CG(R)4 #050783 was calibrated in June 2014 and March 2017 at the Physikalisch-Meteorologisches Observatorium Davos/World Radiation Center (PMOD/WRC). Given the two calibration coefficients of the second instrument (see Table 1), we estimate that its degradation is very small, lower than 0.08%/yr."

**Page 5: There is no need for a 3.0.1 title here.**

Authors: Don